# Engineering of a synthetic quadrastable gene network to approach Waddington landscape and cell fate determination

Fuqing Wu[1], Ri-Qi Su[1,2], Ying-Cheng Lai[2,3,4], Xiao Wang[1]*

[1]School of Biological and Health Systems Engineering, Arizona State University, Tempe, United States; [2]School of Electrical, Computer and Energy Engineering, Arizona State University, Tempe, United States; [3]Institute for Complex Systems and Mathematical Biology, King's College, University of Aberdeen, Aberdeen, United Kingdom; [4]Department of Physics, Arizona State University, Tempe, United States

**Abstract** The process of cell fate determination has been depicted intuitively as cells travelling and resting on a rugged landscape, which has been probed by various theoretical studies. However, few studies have experimentally demonstrated how underlying gene regulatory networks shape the landscape and hence orchestrate cellular decision-making in the presence of both signal and noise. Here we tested different topologies and verified a synthetic gene circuit with mutual inhibition and auto-activations to be quadrastable, which enables direct study of quadruple cell fate determination on an engineered landscape. We show that cells indeed gravitate towards local minima and signal inductions dictate cell fates through modulating the shape of the multistable landscape. Experiments, guided by model predictions, reveal that sequential inductions generate distinct cell fates by changing landscape in sequence and hence navigating cells to different final states. This work provides a synthetic biology framework to approach cell fate determination and suggests a landscape-based explanation of fixed induction sequences for targeted differentiation.

*For correspondence: xiaowang@asu.edu

## Introduction

Multistability is a mechanism that cells use to achieve a discrete number of mutually exclusive states in response to environmental inputs, such as the lysis/lysogeny switch of phage lambda (*Arkin et al, 1998*; *Oppenheim et al., 2005*) and sporulation/competence in *Bacillus subtilis* (*Süel et al., 2006*; *Schultz et al., 2009*). In multicellular organisms, multistable switches are also common in the cellular decision-making including the regulation of cell-cycle oscillator during cell mitosis (*Pomerening et al., 2003*), Epithelial-to-Mesenchymal transition and cancer metastasis (*Jolly et al., 2016*; *Lee et al., 2014a*), and the well-known cell differentiation process, which is a manifestation of cellular state determination in a multistable system (*Laurent and Kellershohn, 1999*; *Guantes and Poyatos, 2008*). However, loss of multistability can drive cells to acquire metastatic characteristics and stabilize highly proliferative, pathogenic cellular states in cancer (*Lee et al., 2014b*).

C.H. Waddington hypothesized the 'epigenetic landscape' to explain canalization and fate determination mechanism during cell differentiation (*Waddington, 1957*). In this hypothesis, differentiation is depicted as a marble rolling down a landscape with multiple bifurcating valleys and eventually settles at one of the local minima, corresponding to terminally differentiated cells. More recent theoretical studies further proposed the local minima to be modeled as steady states or attractors of dynamical systems, which can be mathematically described using differential equations (*Zhang and Wolynes, 2014*; *Li and Wang, 2013a*). As such, cell differentiation can be interpreted as a state transition process on a multistable dynamic system. A myriad of theoretical analysis have

**eLife digest** Cells in animals use a process called differentiation to specialize into specific cell types such as skin cells and liver cells. Proteins called transcription factors drive particular steps in differentiation by controlling the activity of specific genes. Many transcription factors interact with each other to form complex networks that regulate gene activity to determine the fate of a cell and control the whole differentiation process. Some individual gene networks can program cells to become any one of several different cell fates, a feature known as multistability.

In the 1950s, a scientist called Conrad Waddington proposed the concept of an "epigenetic landscape" to describe how the fate of a cell is decided as an animal develops. The cell, depicted as a ball, rolls down a rugged landscape and has the option of taking several different routes. Each route will eventually lead to a distinct cell fate. As the ball moves down the hill, the choice of routes and final destinations becomes more limited. Theoretical approaches have been used to understand how gene regulatory networks shape the epigenetic landscape of an animal. However, few studies have experimentally tested the findings of the theoretical approaches and it is not clear how environmental inputs help to determine which path a cell will take.

Although bacteria cells do not generally specialize into particular cell types, bacteria cells can use multistability in transcription factor networks to switch between different behaviors or "states" in response to cues from the environment. Wu et al. used a bacterium called *E. coli* as a model to investigate how a gene network called MINPA from mammals, which is involved in differentiation and is believed to show multistability, can guide cells to adopt different states. The work combined experimental and mathematical approaches to design, construct and test an artificial version of the MINPA gene network in *E. coli*.

The experiments showed that MINPA could direct the cells to adopt four different stable states in which the cells produced fluorescent proteins of different colors. With the help of mathematical modeling, Wu et al. charted how the landscape of cell states changed when external chemical cues were applied. Exposing the cells to several cues in particular orders guided the cells to different final states.

The findings of Wu et al. shed new light on how the fate of a cell is determined and provide a theoretical framework for understanding the complex networks that control cell differentiation. This could help develop new ways of directing cell fate that could ultimately be used to generate cells to treat human diseases.

investigated the functioning of such systems and quantified the Waddington landscape and developmental paths through computation of the probability landscape for the underlying gene regulatory networks (*Li and Wang, 2013a*; *Wang et al., 2011*; *Li and Wang, 2013b*; *Ferrell, 2012*; *Bhattacharya et al., 2011*; *Macarthur et al., 2009*; *Huang et al., 2007*). Recent studies also revealed that the potential landscape and the corresponding curl flux are crucial for determining the robustness and global dynamics of non-equilibrium biological networks (*Wang, 2015*; *Xu et al., 2014*; *Wang et al., 2008*). Furthermore, the multiple stable steady states have been predicted beyond the bistable switches with or without epigenetic effects, which is reflected in slow timescales (*Wang, 2015*; *Xu et al., 2014*; *Li and Wang, 2013b*; *Feng and Wang, 2012*; *Wang et al., 2011*; *Feng et al., 2011*). Experimental researches, however, mostly focus on bistable switches, involving transitions between only two states. And demonstrations, from a combination of experiments and computational modeling, for the existence and operation of such a landscape in a higher dimensional multistable system are still lacking. Moreover, it remains unknown how gene regulatory networks (GRNs), gene expression noise, and signal induction together shape the attractor landscape and determine a cell's developmental trajectory to its final fates (*Schmiedel et al., 2015*; *Tanouchi et al., 2015*; *Prindle et al., 2014*; *Chalancon et al., 2012*; *Murphy et al., 2010*; *Balázsi et al., 2011*; *Kramer and Fussenegger, 2005*; *Bennett et al., 2008*; *Maamar et al., 2007*).

Complex contextual connections of GRNs have impeded experimentally establishing the shape and function of the cell fate landscape. Rationally designed and tunable synthetic multistable gene networks in *E. coli*, however, could form well-characterized attractor landscapes to enable close

experimental investigations of general principles of GRN regulated cellular state transitions. Since the functioning of these principles only requires the most fundamental aspects of gene expression regulation, they would also be applicable for cell differentiation regulations in mammalian cells. Here, we combine mathematical theory, numerical simulations, and synthetic biology to probe all possible sub-networks of mutually inhibitory network with positive autoregulations (MINPA, *Figure 1A*), which has been hypothesized to have multistability potentials (*Guantes and Poyatos, 2008*; *Huang et al., 2007*). Moreover, MINPA and its sub-networks are recurring motifs enriched in GRNs regulating hematopoietic development (Gata1-Pu.1, [*Graf and Enver, 2009*]), trophectoderm differentiation (Oct3/4-Cdx2, [*Niwa et al., 2005*]), endoderm formation (Gata6-Nanog, [*Bessonnard et al., 2014*; *Li and Wang, 2013a*]), and bone, cartilage, and fat differentiation (RUNX2-SOX9-PPAR-γ, [*MacArthur et al., 2008*; *Rabajante and Babierra, 2015*]).

## Results

### MINPA circuit construction and multistability analysis

Engineered circuits of MINPA (*Figure 1B*) and its sub-networks (*Figure 1—figure supplement 1A*) are designed to use two hybrid promoters, *Para/lac* and *Plux/tet,* which are characterized experimentally to show small leakage and high nonlinearity (*Figure 1D–E* and *Figure 1—figure supplement 1B–D*). For MINPA topology, hybrid promoter *Para/lac* drives *AraC* and *TetR* expression, representing the node X in *Figure 1A*, whereas *Plux/tet* controls *LuxR* and *LacI* transcription, representing the node Y. AraC and LuxR activate *Para/lac* and *Plux/tet* in the presence of Arabinose and AHL (3oxo-C6-HSL) respectively, forming positive autoregulations. IPTG inhibits the repressive effect of LacI on TetR expression, while aTc counteracts TetR repression on LacI. Hence, the two nodes form the topology presented in the conceptual design shown in *Figure 1A*. Green fluorescent protein (GFP) and mCherry serve as the corresponding readouts of *Plux/tet* and *Para/lac* activities in living cells (*Figure 1B*).

Topologies of MINPA and all its subnetworks can be divided into four layers, from one- to four-dimensional networks based on the number of regulatory edges (*Figure 1C* and *Figure 1—figure supplement 1E–F*) and further categorized into nine groups based on the configurations of activation and inhibition. By computationally searching a large parameter range for each of the nontrivial networks (*Faucon et al., 2014*), we found that networks with two auto-activations, including $A^2$, $RA^2$, $R^2A^2$, have high probability of tristability or quadrastability (*Figure 1—figure supplement 1G*), defined as having three or four stable steady sates (SSS) under a common induction condition. However, MINPA has broader parameter distributions than the other two (*Figure 1—figure supplement 1H–I*), which suggests it is more resistant to parameter change and thus likely to achieve multistability in experimental settings.

### Systematical multistability evaluation of MINPA and its sub-networks

In order to experimentally evaluate dynamic properties of these networks, we constructed nine circuits including tunable positive feedbacks (T6 and T9), mutual inhibition (T5), dual-positive feedbacks (T10), and their combinations (T7, T11, T13, T14 and T15). One-dimensional networks (T1, T4, T2 and T8) and trivial two-dimensional networks (T3 and T12) are excluded for their low multistability probability. All motifs were constructed using the same set of components (*Figure 1*).

Probing a circuit's multistability typically requires thorough hysteresis experiments covering wide ranges of doses for all inducers (*Acar et al., 2005*; *Angeli et al., 2004*; *Gardner et al., 2000*), which becomes infeasible for nine complex networks with four inducers. To improve the efficiency of probing multistability and tunability, we designed a 'sequential induction' method to accelerate exploration of unknown high dimensional bifurcation spaces (see Appendix text for details), instead of conventional 'back and forth' hysteresis on one parameter dimension. The main concept relies on the fact that multistable gene networks could exhibit discontinuous jump from one state to another in response to changing parameter (inducer) combinations. Taking the classic 'toggle switch' as an example (*Gardner et al., 2000*), the circuit can be tuned by two external inducers and its two-parameter bifurcation diagram has a stretched *S* shape (*Figure 2A*). Initialized at an arbitrary state A, the cells could reach State C in the bistable region directly when induced with both inducers simultaneously. If the cells are first induced by Inducer I to go to state B, they will also reach State C after Inducer II is added. However, if the same dose of Inducer II is applied first, cells will cross the

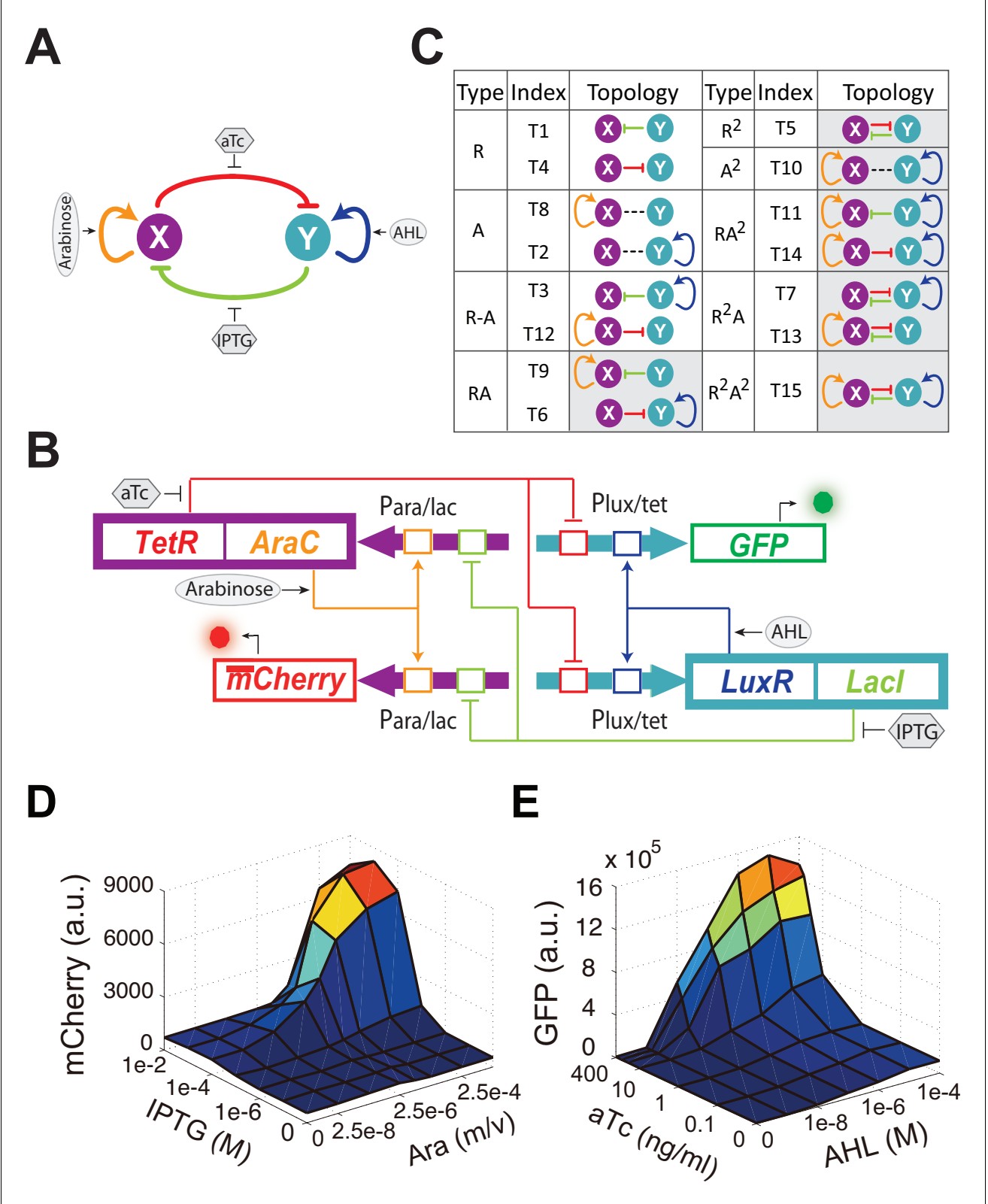

**Figure 1.** Conceptual and experimental design of MINPA and its sub-networks. (**A**) Abstract diagram of MINPA topology, where X and Y mutually inhibit (T-bars) each other and auto-activate (arrowheads) itself. Four inducers to regulate the four color-coded regulatory edges are also listed. (**B**) Molecular implementation of the MINPA network. *Para/lac* (purple arrow) is activated by AraC (yellow) and repressed by LacI (light green), while *Plux/tet* (cyan arrow) is activated by LuxR (blue) and repressed by TetR (red). Arabinose and AHL (oval) can induce AraC and LuxR activation, respectively.

*Figure 1 continued on next page*

*Figure 1 continued*

IPTG and aTc (hexagon) can respectively relieve LacI and TetR inhibition. GFP and mCherry serve as the readout of *Para/lac* and *Plux/tet*, respectively. Therefore, TetR and AraC collectively form the node X in (**A**), color-coded as purple rectangle. Similarly, LuxR and LacI collectively form the node Y in (**A**), color-coded as cyan rectangle. Genes, promoters and regulations are color-coded corresponding to the topology in (**A**). (**C**) List of MINPA and its 14 sub-networks. Numbering of indices is converted from topologies' binary name (see *Figure 1—figure supplement 1E* for more details). T represents 'topology'. R represents 'repression', and A represents 'autoactivation'. Superscript is used to describe the number of such types of edges. Topologies with shaded background were later constructed and analyzed experimentally. (**D–E**) Dynamic responses for *Para/lac* (**D**) and *Plux/tet* (**E**) through induction with Arabinose (Ara) and IPTG, and AHL and aTc, respectively. Presented data was the mean value of three replicates. mCherry and GFP serves as the readout of the two promoters.

The following figure supplement is available for figure 1:

**Figure supplement 1.** Experimental design, topological hierarchy and multistability probability analysis of MINPA sub-networks.

bifurcation plane to state D on the low-Response surface and then reach state E with addition of Inducer I (*Figure 2A*). State C and E are two different steady states with the same induction dosages, illustrating hysteresis and verifying multistability.

To test our theoretical analysis, a synthetic toggle switch circuit was constructed (*Figure 2—figure supplement 1A*). Following experimental design principles (see Appendix text for details), we designed a protocol to show the sequential induction effects. We first employed IPTG to induce the circuit for 5 hr, and then aTc was added. Time course results showed that cells stayed at low-GFP state till 24 hr (*Figure 2—figure supplement 1B*). However, cells induced with aTc first, and then IPTG mainly stayed at high-GFP state, another stable steady state under this condition. Simultaneous aTc and IPTG induction produced similar cell distributions. These results show that sequential induction can be used as a strategy to quickly explore a multistable potential landscape for complex non-equilibrium systems.

Without knowing the exact bifurcation range beforehand, such ordered sequential inductions could help quickly explore the irregular bifurcation space to reveal multistability for systems with complicated bifurcations, which is typically caused by interfering parameters. Similar sequential induction techniques have been shown to enable access of otherwise hard-to-reach cell death states in breast cancer cells (*Lee et al., 2012*). This strategy has also been widely employed in directed differentiation of stem cells to specific lineages (*Paşca et al., 2015*; *Pagliuca et al., 2014*; *Kroon et al., 2008*) and reprogramming somatic cells to induced pluripotent stem cells (*Liu et al., 2013*). Although specific inducer concentrations are required to observe the effects of this strategy in synthetic circuits, sequential induction with pre-selected inducer combinations can help perform a coarse-grained exploration from different directions in the parameter space. Furthermore, stochastic gene expression of the circuits also contributes to cellular population distribution thus leads to pronounced sequential induction effects, given experimentally feasible amount of time, when the system is entering its multistable region from different directions. Therefore, distinct final states, or even different population distributions, under sequential induction strongly suggests the existence of nonlinear dynamics, including multistability (see Appendix text for details).

Using the sequential induction approach, we tested the nine circuits using flow cytometry. Cells were first induced by inducer I, inducer II was then added into the media for another 24 hr. Depending on the network configuration, four different dual-inducer combinations were used. For example, Arabinose and IPTG were applied sequentially and simultaneously to T9, T13, T11 and T15, respectively (*Figure 2B*). It can be seen only T15 exhibits significant expression difference between three induction patterns, while the others show little change (*Figure 2B* and *Figure 2—figure supplement 2A*). It should be noted that T15 also exhibits tri-modality of fluorescence expression, suggesting multistability given the presence of gene expression noise, which is partially consistent with our computational predictions. Similarly, AHL and aTc were applied to T6, T7, T14, and T15, respectively (*Figure 2C* and *Figure 2—figure supplement 2B*). Results show that only T15 exhibits significant fluorescence pattern change with different inductions, whereas T6 and T7 exhibit minor uniform shifts of expression. T14, although exhibiting bimodality, only shows a ratio change of two populations between three inductions and no sign of bifurcation. Sequential induction by Arabinose and AHL combinations has little effect on T10, T14 and T11, but T15 displays three notable populations

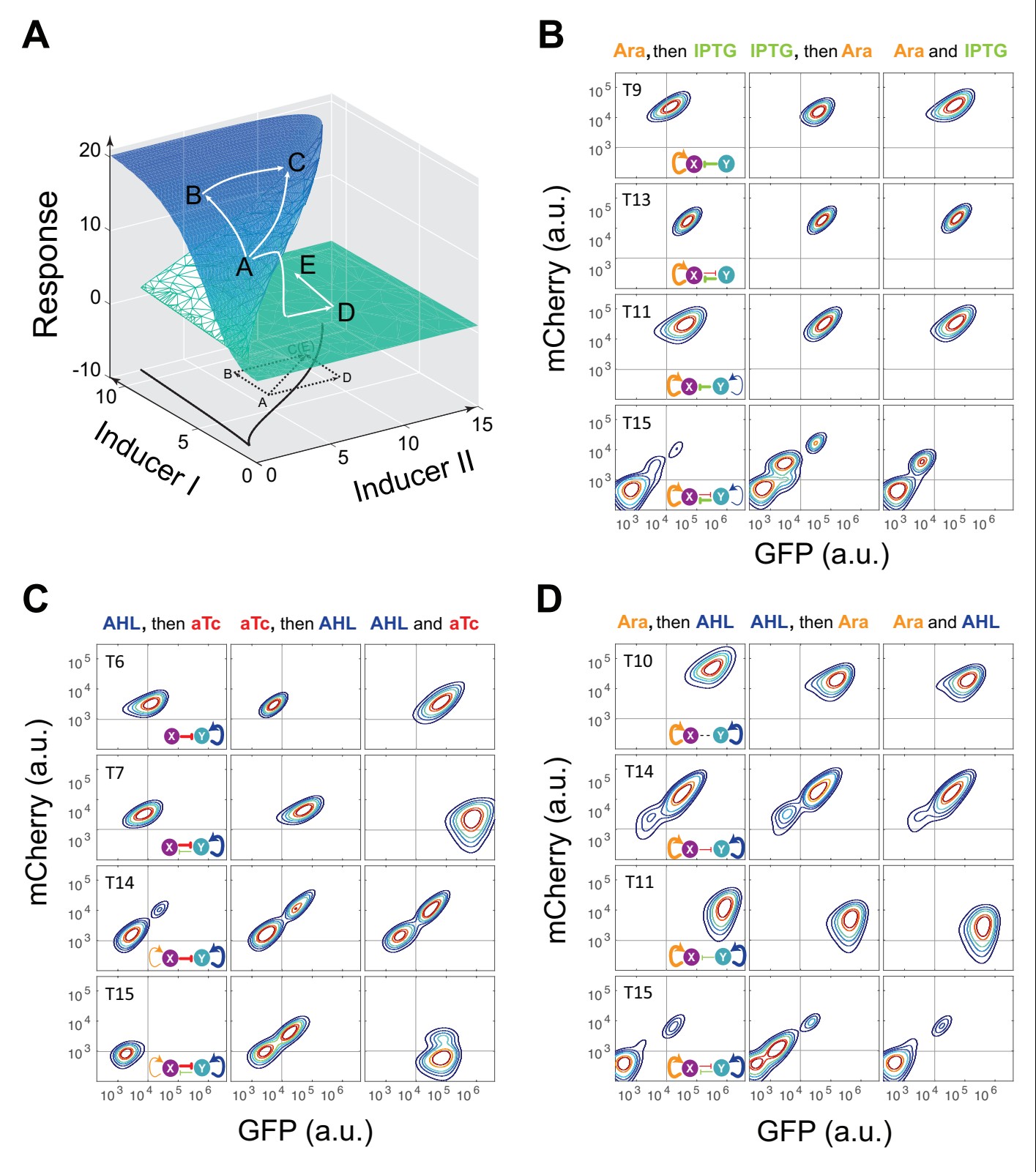

**Figure 2.** Sequential induction of MINPA and its sub-networks. (**A**) Schematic illustration of rationale for sequential induction. This two-parameter bifurcation diagram of a bistable toggle-switch depicts all steady state values of response (Z-axis) with combinations of inducer I and II (X and Y axes). Arrows illustrate order and direction of inductions and consequent steady state value changes. Solid lines on the X-Y plane are the boundaries of bistability. Dashed lines on the X-Y plane are projections of solid white arrowheads. (**B**) Arabinose (Ara) and IPTG were sequentially (left and middle

*Figure 2 continued on next page*

*Figure 2 continued*

columns) or simultaneously (right column) applied to induce T9, T13, T11, and T15. T: topology. The concentration of Arabinose and IPTG is $2.5*10^{-5}$m/v, and $5*10^{-5}$ M, respectively. To indicate the effects of inducers, we used the same color for applied inducers and its regulated connections, which were also shown in bold lines. The other non-regulated connections are represented by thin lines. (C) AHL and aTc were sequentially (left and middle) or simultaneously (right) applied to induce T6, T7, T14, and T15. The concentration of AHL and aTc is $1*10^{-4}$ M, and 200 ng/ml, respectively. (D) Ara and AHL were sequentially (left and middle) or simultaneously (right) applied to induce T10, T14, T11, and T15. The concentration of Arabinose and AHL is $2.5*10^{-5}$m/v, and $1*10^{-8}$ M, respectively. Samples were treated with the first inducer till $OD_{600}$ is about 0.15 and then the second inducer was added. Cells were grown for another 24 hr before measured by flow cytometry. The experiments were performed in triplicate and repeated two times, and representative results are presented. The inducers are color-coded as visual assistance to indicate which edge of inset diagram it regulates.

The following figure supplements are available for figure 2:

**Figure supplement 1.** Experimental design and validation of sequential induction strategy in a synthetic toggle switch circuit.

**Figure supplement 2.** Time course results of sequential induction for the MINPA (**T15**) circuit.

**Figure supplement 3.** Sequential induction for circuits T5, T7, T13, and T15 with inducers IPTG and aTc.

for AHL-then-Arabinose induction (*Figure 2D* and *Figure 2—figure supplement 2C*). IPTG and aTc were also tested on T5, T7, T13 and T15, but no notable dynamics were observed (*Figure 2—figure supplement 2D* and *Figure 2—figure supplement 3*). Taken together, T15, the full MINPA topology, shows the most variety and complexity in population heterogeneity under sequential inductions, suggesting this circuit has the highest potential to generate complex multistability within our induction range and hence enable us to approach the Waddington landscape.

## Bifurcation and hysteresis verification of multistability

Next, operating principles and full tunability of T15 (MINPA) were further examined by using four inducers (Arabinose, AHL, aTc, and IPTG) to fine tune the strength of regulations and perturb the system (*Figure 3A*). Uninduced cells showed low GFP and low mCherry expression (low-low state, LL). In the presence of AHL and aTc, high GFP and low mCherry (GFP state) is observed; low GFP and high mCherry (mCherry state) emerged with induction of Arabinose; and high GFP and high mCherry (high-high state, HH) was achieved when induced with Arabinose and AHL. These results verify that our engineered MINPA circuit is functioning as designed and fully controllable with four distinct states reachable through appropriate inductions, respectively.

To help design experiments to further investigate the circuit's quadrastability, a detailed mathematical model was developed to describe the system (see Appendix for details). Using parameters derived from hybrid promoter testing experiments, bifurcation analysis was carried out to systematically quantify MINPA's dynamic behavior (*Figure 3B*, *Figure 3—figure supplement 1* and *Figure 3—figure supplement 2A–H*). *Figure 3B* is the three-dimensional bifurcation diagram, where levels of GFP and mCherry represent the states of node X and Y, and 'AR/AL' is a lumped parameter composed of a fixed ratio of the concentrations of Arabinose and AHL. Overall, it can be seen that the system, initialized without induction, is predicted to be quadrastable (shown as four colored spheres, representing LL (grey), GFP (green), mCherry (rose), and HH (golden) state, respectively) but with the low-low state to have dominant attractiveness (shown as the big gray sphere) when AR/AL is low (C1). However, when AR/AL level is within an intermediate range, relative stabilities between different states become comparable. When AR/AL level increased from C1 to C2, the circuit's quadrastability becomes well pronounced, illustrated as four similar-sized colored spheres on the same gray plane, which represents the low-low, GFP, mCherry, and high-high state, respectively (*Figure 3—figure supplement 1*). As AR/AL continues to increase from C2 to C3, while the other three SSS remain stable, the stability of the GFP branch disappears. Further increase of AR/AL results in only one stable state-the high-high state, shown as the orange sphere with biggest size.

To establish MINPA's quadrastability and tristability as predicted, hysteresis, a hallmark of multistability (*Acar et al., 2005*; *Wu et al., 2014*, *2013*), of the network was tested. Initialized at the low-low state, cells were induced by increasing doses of AR/AL corresponding to C1 to C4 and measured by flow cytometry (*Figure 3C* and *Figure 3—figure supplement 2I*). As predicted, $C1_{LL}$ (cells

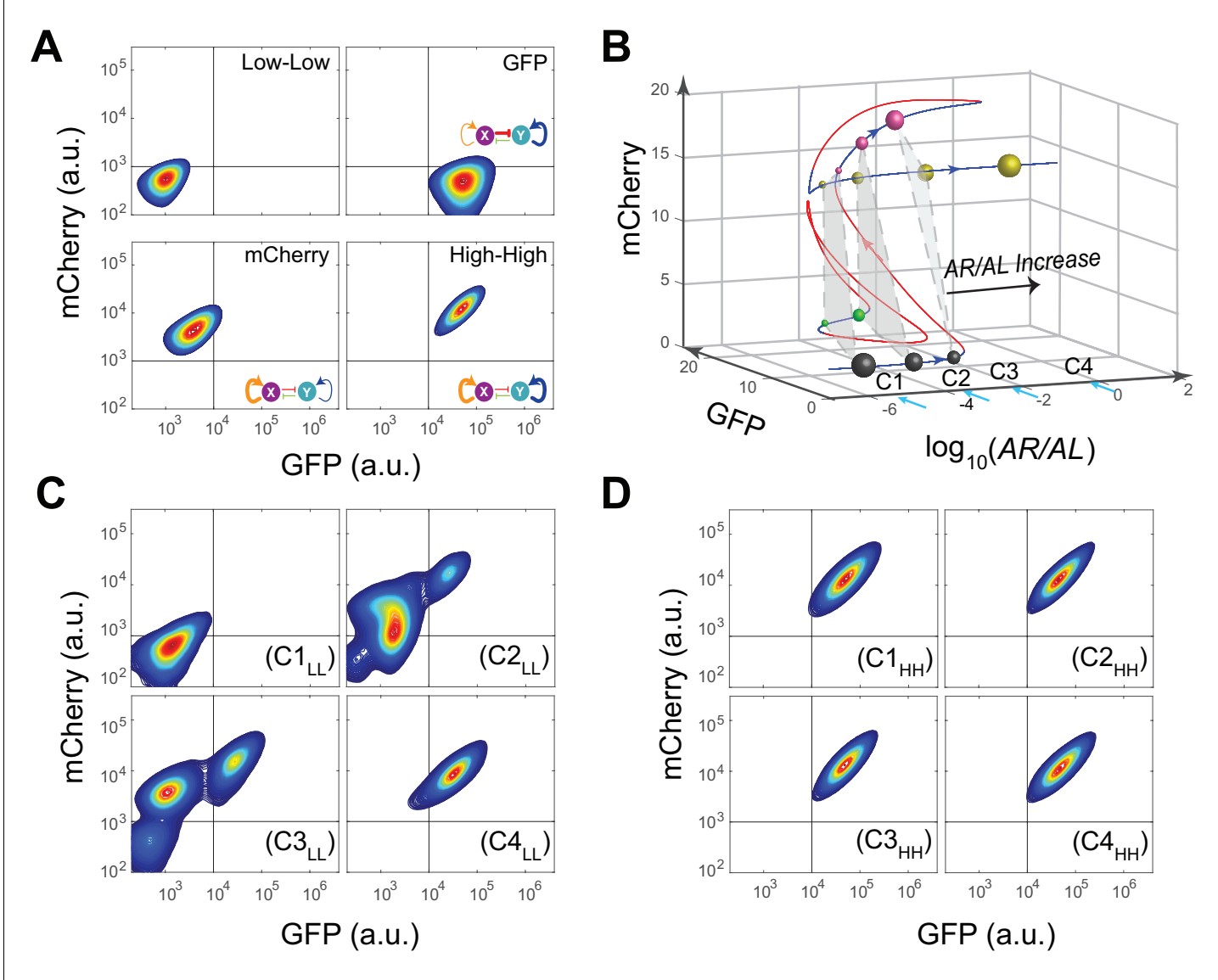

**Figure 3.** Bifurcation analysis and hysteresis of MINPA. (**A**) Engineered MINPA is tunable to reach four individual states: low-low, GFP, mCherry, and high-high, under no induction, $1*10^{-4}$ M AHL and 100 ng/ml aTc, $2.5*10^{-5}$ (m/v) Arabinose, $1*10^{-4}$ M AHL and $2.5*10^{-3}$ (m/v) Arabinose, and respectively. To indicate the effects of inducers, we used the same color for applied inducer and its regulated connection (bolder lines) in the MINPA topology. The other non-regulated connections are represented by thin lines. (**B**) 3-D bifurcation diagram of MINPA. *AR/AL* is a lumped parameter composed of increasing concentrations of Arabinose and AHL, but the ratio of Arabinose and AHL is fixed, i.e., [Arabinose]/[AHL] is a constant. GFP and mCherry represent the states of node X and Y. Blue lines represent stable steady states, while red ones are unstable steady states. Grey, green, rose, and golden spheres represent low-low, GFP, mCherry, and high-high state, respectively. And the size of spheres correlates with the attractiveness of each state. C1, C2, C3, and C4 are four increasing concentrations of Arabinose and AHL used for experimental probing. (**C–D**) Hysteresis results of MINPA under induction of *AR/AL*. C1$_{LL}$-C4$_{LL}$: cells with low-low initial state (**C**) are induced with *AR/AL* at C1 to C4; C1$_{HH}$-C4$_{HH}$: cells with high-high initial state (**D**) are induced with *AR/AL* for 24 hr at C1 to C4. C1: no inducers; C2: $2.5*10^{-6}$m/v Arabinose and $1*10^{-7}$ M AHL; C3: $2.5*10^{-5}$m/v Arabinose and $1*10^{-6}$ M AHL; C4: $2.5*10^{-3}$m/v Arabinose and $1*10^{-4}$ M AHL. Arabinose and AHL were added at the same time to induce the cells. 100,000 cells were recorded for each sample by flow cytometry.

The following figure supplements are available for figure 3:

**Figure supplement 1.** Another view of the 3-D bifurcation diagram of MINPA at C2.

**Figure supplement 2.** Bifurcation analysis for and hysteresis of MINPA with induction of Arabinose and AHL.

with initial Low-Low state grown at C1 condition) experiment demonstrates uniform low-low fluorescence profile, due to the low-low state's dominant attractiveness, and C4$_{LL}$ shows a uniform high-high profile. Interestingly, C3$_{LL}$ indeed illustrates tri-modality, which is the result of predicted tristability. C2$_{LL}$ experiment, on the other hand, exhibits enough heterogeneity to signal high-high, low-low, and mCherry state, but does not illustrate significant trace of GFP state. Given that GFP state is achieved through combinational induction of AHL and aTc (*Figure 3A*), we hypothesize that the GFP state here is not easily accessible with AHL induction only. Next, cells initialized at high-high states were collected and diluted into fresh media with the same concentrations of *AR/AL* (*Figure 3D* and *Figure 3—figure supplement 2J*). As predicted, these cells keep high-high expression profile even with inductions as low as C1, another demonstration that the system is already multistable at C1. Taken together, the two sets of experiments demonstrated clear hysteresis and verified the existence of three of the four predicted SSS.

## Experimental demonstration of model-guided quadrastability of MINPA

To further investigate what determines the accessibility of certain SSS in this quadrastable system and how cells navigate this attractor landscape, we take into account gene expression stochasticity (*Wu et al., 2013*) to sketch out MINPA's quasi-potential attractor landscape (*Figure 4A* and Appendix), which is calculated as the negative logarithmic function of stationary distribution density in the phase space of GFP and mCherry. Using the weighted ensemble random walk algorithm (Appendix), the stationary density distribution can be efficiently calculated from the initial uniform distribution. It can be seen that when there is no inducer, MINPA is already quadrastable with four local minima, which is consistent with bifurcation analysis for C1 condition. Furthermore, the much stronger stability of the low-low state (deepest well, Top landscape) and high state-transition barrier explain homogeneous low-low population (C1 experiment in *Figure 3C*) when cells were initialized with no inductions.

Since Arabinose and AHL combination is not sufficient to enable the cells to reach all four SSS, we chose to add aTc to the mix to further facilitate cell transitions among these four SSS. Using our expanded model, we simulated simultaneous and sequential inductions and computed corresponding quasi-potential landscape (*Figure 4A*), showing cells harboring the same MINPA network exhibiting distinct landscapes under different inductions. AHL and aTc promote a more stable GFP state (Left center), while Arabinose induction modulates the landscape to be biased toward mCherry state (Right center). When the three inducers were applied simultaneously, the landscape changes and the four states show comparable stabilities (Bottom), suggesting a higher possibility of quadramodal cell population experimentally. Experimental validation is shown as flow cytometry measurements of cells treated with Arabinose, AHL, and aTc simultaneously for 24 hr (*Figure 4B*, and *Figure 4—figure supplement 1A*). Such a hybrid induction greatly facilitates the cells' transition from low-low state to the other three states so that a quadramodal distribution emerges. Single-cell time lapse microscopy results also showed that the initial low-low state cells could differentiate into GFP, mCherry and high-high state cells (*Figure 4—figure supplement 1B–D* and *Appendix 1—Video 1*). This also finally verifies predicted quadrastability of MINPA.

There are two other strategies to reach this condition: sequential inductions with AHL-and-aTc and then Arabinose (*Figure 4A*, Left route) or Arabinose and then AHL-and-aTc (Right route). Even though the initial and final landscapes are the same, the dynamics for each route are quite different, which could lead to distinct outcomes. By comparing state barrier heights (*Figure 4A*), we hypothesize that cells walking through the left route would start transitioning from low-low state to GFP state upon induction of AHL and aTc. Following Arabinose induction would then make the mCherry state accessible. So some cells with GFP state would transition to high-high state while some low-low state cells transition to mCherry state, resulting in cells in all four states. Experimental testing indeed shows four stable populations (*Figure 4C*). At 6.5 hrs of AHL and aTc induction, about 12% cells were moving to GFP state while the rest of them still stay 'undecided' at low-low state (*Figure 4—figure supplement 1A and E*). This is consistent with the simulated landscape as these two states are more stable and accessible to each other (*Figure 4A*, Left). Arabinose induction promoted some cells to transition into mCherry state while some cells continued moving into GFP state, of which some further transitioned to high-high state.

Interestingly, the right route is predicted to generate different results. When first induced with Arabinose, the mCherry valley is so deep that it would be difficult for cells to jump out to high-high state, and low-low state cells are also hardly transit to GFP state due to its low attractiveness, and

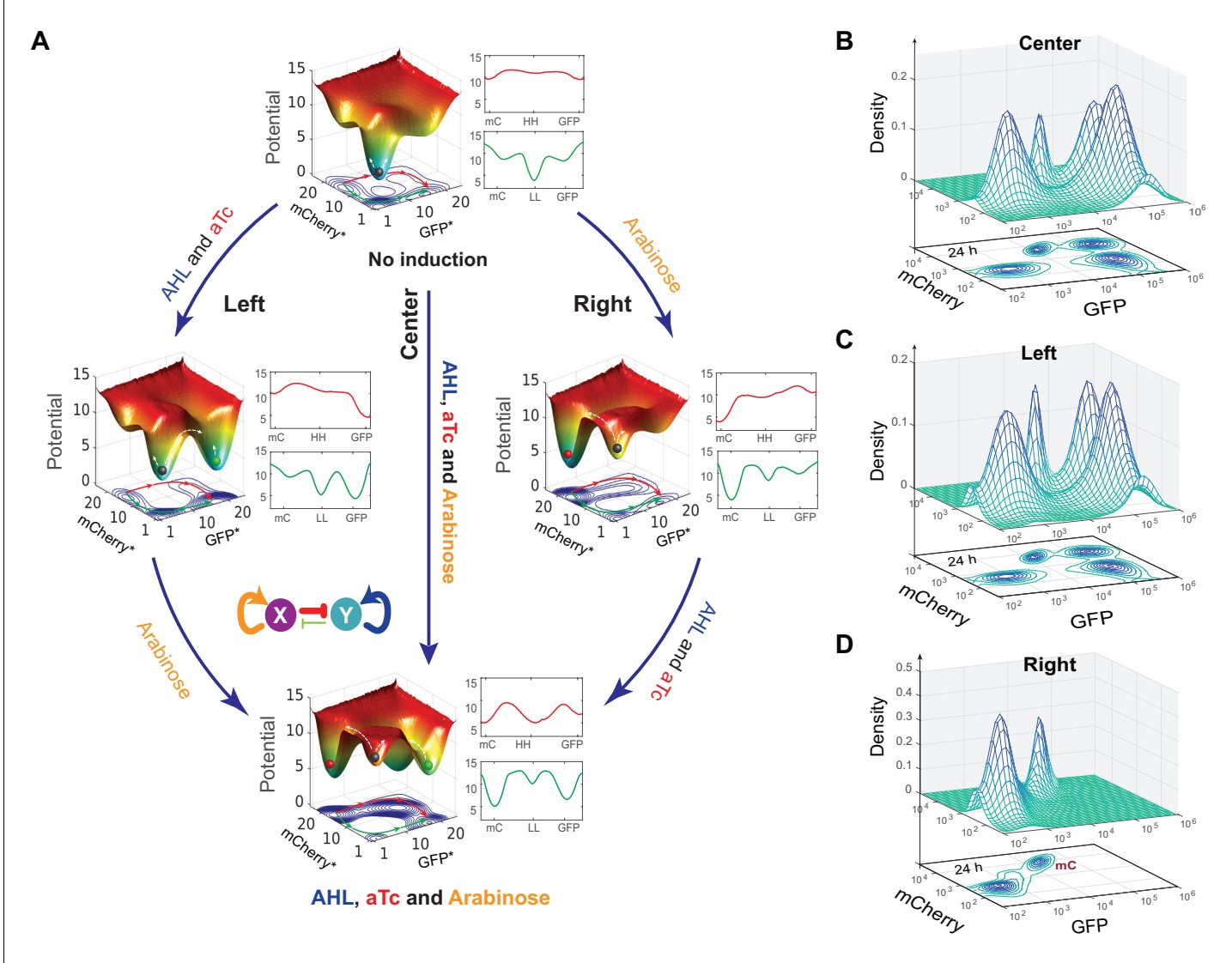

**Figure 4.** Model-guided quadrastability of MINPA through triple induction. (**A**) Dynamic evolution of computed energy landscapes of MINPA under sequential/simultaneous inductions of Arabinose, and/or AHL and aTc. Center route: simultaneous induction with three inducers; Left route: sequential induction with AHL and aTc first, and then Arabinose. Right route: sequential induction with Arabinose, and then AHL and aTc. Deeper wells represent the higher stability of corresponding states. For each three-dimensional landscape, corresponding two-dimensional state-potential plots were also shown. Red line sketches the potentials from mCherry state to high-high to GFP state while green one represents the potentials from mCherry state to low-low to GFP states. mC: mCherry; HH: high-high; LL: low-low. GFP* and mCherry* is the computed GFP and mCherry abundance from the model. To indicate the effects of inducers, we used the same color for applied inducers and its regulated connections, which were also shown in bolder lines. (**B–D**) Experimental validations of model-predicted quadrastability using flow cytometry. Quadrastable steady states were observed when Arabinose, AHL, and aTc were simultaneously added into the media (**B**), corresponding to the Center route in A). Four populations were also observed when AHL and aTc were first added to growth media for 6.5 hr and then Arabinose was added, and cells were grown for another 24 hr before measurement (**C**), corresponding to the Left route in **A**). Bimodality (low-low and mCherry states) was generated when Arabinose was first applied and then AHL and aTc were added (**D**), corresponding to the Right route in **A**). Concentrations for Arabinose, AHL and aTc are 2.5*10$^{-5}$m/v, 1*10$^{-4}$ M, and 400 ng/ml, respectively. Representative results from three replicates are showed and 100,000 cells were recorded for each sample by flow cytometry.

The following figure supplement is available for figure 4:

**Figure supplement 1.** Cells' states under induction with the first inducer, microfludic results to demonstrate quadrastability with IPTG and aTc induction, and time course of sequential induction of AHL, aTc and Ara.

thus most cells would stay at mCherry and low-low state even with AHL and aTc inductions (*Figure 4A*, Right). Experimental testing of the right route indeed only produces two populations with low-low and mCherry state (*Figure 4D*). With 5 hrs of Arabinose induction, most cells still stay at low-low state because of slow transition to the mCherry state (*Figure 4—figure supplement 1A*), but 84.6% cells transitioned to mCherry state with 15.3% cells at low-low state at 9.5 hr (*Figure 4—figure supplement 1A*). This is consistent with our model predictions. The high barrier between the mCherry state and high-high state blocks the transition from mCherry state to high-high state, while the low attractiveness and relatively high barrier of the GFP state also decreases the probability of cells transitioning from low-low to GFP state. Hence, when AHL and aTc are applied, cells are predominantly in the mCherry state with a small portion in low-low state with low probability of transitioning out, resulting in a bimodal distribution.

## Discussion

Multistability and the resulting landscape has long been proposed as an underlying mechanism that cells use to maintain pluripotency and guide differentiation (*Kauffman, 1993*; *Laurent and Kellershohn, 1999*; *Huang et al., 2007*; *Guantes and Poyatos, 2008*; *Palani and Sarkar, 2009*; *Narula et al., 2010*; *Faucon et al., 2014*). Theoretical frameworks have also been established to quantify the Waddington landscape and biological paths for cell development (*Li and Wang, 2013a*, *2013b*; *Wang et al., 2011*). Experimental validation of this hypothesis and a full understanding of this mechanism will help reveal differentiation dynamics and routes for all cell types, which remains an outstanding problem in biology.

In this study, we engineered the quadrastable MINPA circuit and show that it can guide cell fate choices, represented by fluorescence expression, through shaping the potential landscape. MINPA represents one of the most complicated two-node network topologies and includes four genes to implement a web of regulations. Biological complexity correlates with the number of regulatory connections (*Szathmáry et al., 2001*), not the number of genes. Hence, dense connectivity and complex dynamics of MINPA may provide a framework to understand similarly densely connected gene regulatory networks.

Combining mathematical modeling and experimental investigation, this study serves as a proof-of-principle demonstration of the Waddington landscape. Furthermore, we used this circuit to demonstrate how different sequential inductions can change the landscape in a specific order and navigate cells to different final states. Such illustrations suggest mechanistic explanations of the need for fixed induction sequences for targeted differentiation to desired cell lineage. Overall, this study helps reveal fundamental mechanisms of cell-fate determination and provide a theoretical foundation for systematic understanding of the cell differentiation process, which will lead to development of new strategies to program cell fate.

## Materials and methods

### Strains, Media, and Chemicals

All the molecular cloning experiments were performed in *E.coli* DH10B (Invitrogen, USA), and measurements of MINPA and sub-networks were conducted in *E.coli* K-12 MG1655$\Delta$*lacI*$\Delta$*araCBAD* strain as previously described (from Dr. Collins Lab [*Litcofsky et al., 2012*]). The sequential induction for the toggle circuit was conducted in *E.coli* MG1655$\Delta$*lacI* strain as previously described (*Litcofsky et al., 2012*). Cells were grown at 37°C in liquid and/or solid Luria-Bertani broth medium with 100 μg/mL ampicillin or kanamycin. Chemicals AHL (3oxo-C6-HSL, Sigma-Aldrich), Arabinose (Sigma-Aldrich, USA), isopropyl $\beta$-D-1-thiogalactopyranoside (IPTG, Sigma-Aldrich), and anhydrotetracycline (aTc, Sigma-Aldrich) were dissolved in ddH2O and diluted into indicated working concentrations. Chemical aTc solution was stocked in brown vials, and experiments involving aTc were performed in cabinet without light, and cell cultures were grown in darken incubator at 37°C. Cultures were shaken in 5 mL and/or 15 mL tubes at 220 rotations per minute (r.p.m).

## Plasmids construction

All the plasmids (MINPA and its nine sub-networks) in this study were constructed using standard molecular cloning protocols and assembled by standardized BioBricks methods based on primary modules (*Table 1*) from the iGEM Registry (www.parts.igem.org). Hybrid promoter *Para/lac* was from Dr. Collins lab and amplified using forward primer: *CGGAATTCGCTTCTAGAGAATTG TGAGCGGATAAC*; and reverse primer: *CGCTGCAGGCACTAGTTTGTGTGAAATTGTTATCCG*. PCR product was purified using GenElute PCR Clean-Up Kit (Sigma-Aldrich), and then cut by restriction enzymes *EcoRI* and *PstI*. The purified product was inserted into pSB1K3 backbone, and finally verified by DNA sequencing. The MINPA circuit was constructed from promoter *Para/lac* and nine other Biobrick standard biological parts: BBa_B0034 (ribosome binding site, RBS), BBa_C0080 (*AraC* gene), BBa_C0040 (*tetR* gene), BBa_K176000 (*Plux/tet* hybrid promoter), BBa_C0062 (*luxR* gene), BBa_C0012 (*lacI* gene), BBa_B0015 (transcriptional terminator), BBa_E0240 (GFP generator), and BBa_J06702 (mCherry generator). The fragment and vector were separated by gel electrophoresis (1% TAE agarose) and purified using GenElute Gel Extraction Kit (Sigma-Aldrich). Then, fragment and vector were ligated together using T4 DNA ligase, and the ligation products were transformed into *E. coli* DH10B and clones were screened by plating on 100 μg/ml ampicillin LB agar plates. Finally, their plasmids were extracted and verified by double digestion (*EcoRI* and *PstI*). The detailed procedures of assembling DNA constructs were described in our previous study (*Wu et al., 2014*). Restriction enzymes (*EcoRI*, *XbaI*, *SpeI*, and *PstI*) and T4 DNA ligase were purchased from New England Biolabs. All the constructs were inserted into high copy number plasmid pSB1A3 and pSB1K3. All the constructs were verified by DNA sequencing (Biodesign sequencing lab in ASU) step by step.

## Flow cytometry

All the samples were analyzed at the indicated time points on an Accuri C6 flow cytometer (Becton Dickinson, USA) with excitation/emission filters (488/530 nm for GFP, and 610 LP for mCherry). The data were collected in a linear scale and non-cellular low-scatter noise was removed by thresholding. All measurements of gene expression were obtained from at least three independent experiments. For each culture, 100,000 events were collected at a slow flow rate. Data files were analyzed using MATLAB (MathWorks).

## Sequential induction and hysteresis

For sequential induction, initially uninduced overnight cell culture was diluted into fresh media without or with inducer I, grown at 37°C and 220 r.p.m till $OD_{600}$ is 0.15 ~ 0.25 (the time usually takes 5 ~ 6.5 hr, depends on the inducers and concentrations). For samples induced individually by Ara, or AHL, or IPTG, it is ~5 hr; for samples induced with aTc, it takes ~6.5 hr. According to our experience,

**Table 1.** Components from the Registry of standard biological parts

| Biobrick number | Abbreviation in the paper | Description |
| --- | --- | --- |
| BBa_C0080 | AraC | AraC arabinose operon regulatory protein from *E. coli* |
| BBa_C0040 | TetR | Tetracycline repressor from transposon Tn10 |
| BBa_C0062 | LuxR | LuxR activator from *Aliivibrio fischeri* |
| BBa_C0012 | LacI | LacI repressor from *E. coli* |
| BBa_E0240 | GFP | GFP generator |
| BBa_J06702 | mCherry | RFP generator |
| BBa_K176002 | $P_{lux/tet}$ | Hybrid promoter with LuxR/HSL- and TetR-binding sites |
| BBa_B0034 | RBS | Ribosome binding site |
| BBa_B0015 | Terminator | Transcriptional terminator (double) |
| BBa_K176009 | CP | Constitutive promoter |
| pSB1K3 | pSB1K3 | High copy BioBrick assembly plasmid with kanamycin resistance |
| pSB1A3 | pSB1A3 | High copy BioBrick assembly plasmid with ampicillin resistance |

gene (GFP) is starting to be partially expressed while steady states are not yet stable. Then inducer II was added into the culture, and grown for another 24 hr. Flow cytometry was performed at 0 hr, 12 hr, and 24 hr after the second inducer was added into the culture. For each set of sequential induction, the first scenario: add inducer I first, then add inducer II; the second scenario: add inducer II first, then add inducer I; the third scenario: add inducers I and II at the same time. As a control, cells without any inducer were also prepared and measured. Inducer I and II were the two of four commercial chemicals: AHL, Arabinose, IPTG, and aTc. All the experiments were repeated for at least three times and only representative results were showed.

For hysteresis experiments, initially uninduced cells were diluted into fresh media and distributed into new 5 ml tubes. Various amounts of Arabinose and AHL (3oxo-C6-HSL) were added into the media, and cells were then grown at 37°C shaker. The initially high-high state cells induced with $2.5*10^{-3}$ m/v Arabinose and $1*10^{-4}$ M AHL were collected with low-speed centrifugation, washed twice, resuspended with fresh medium, and at last inoculated into fresh medium at a 1:100 ratio with the same series of inducer (Arabinose and AHL) concentrations. C1, C2, C3, and C4 (Figure 3B–D) are four increasing concentrations of Arabinose and AHL used for experimental probing, but the ratio of Arabinose and AHL is fixed. Specifically, cells were induced with the Arabinose and AHL at the same time (the third scenario), at concentrations from C1 to C4. C1: no inducers; C2: $2.5*10^{-6}$m/v Arabinose and $1*10^{-7}$ M AHL; C3: $2.5*10^{-5}$m/v Arabinose and $1*10^{-6}$ M AHL; C4: $2.5*10^{-3}$m/v Arabinose and $1*10^{-4}$ M AHL. Flow cytometry analyses were performed at 12 hr and 24 hr to monitor the fluorescence levels. Experiments were repeated two times with three replicates.

## Microfludics and microscopy

Cells with MINPA circuit were grown overnight, which was then re-diluted into 5 mL fresh LB medium with Kanamycin the next day. When $OD_{600}$ of the cells reached about 0.2, cells were spun down with low speed and resuspended in 5 ml of fresh medium and loaded into the device. Detailed description of chip design and device setup could be found from Hasty Lab (Ferry et al., 2011). Two media were prepared: one with inducers and the other without. Cells in the trap were first supplied by the medium without inducer for 6 hr, and then switched to medium with inducers for anther 18 hr, which was controlled by adjusting the heights of the medium syringes relative to one another. Images were taken by using Nikon Eclipse Ti inverted microscope (Nikon, Japan) equipped with an LED-based Lumencor SOLA SE. Phase and fluorescence images were taken every 5 min for 24 hr in total under the magnification 40x. Perfect focus was maintained automatically using Nikon Elements software. Experimental detail can also be found in Appendix.

## Mathematical modeling

Ordinary differential equation models were developed to describe and analyze the MINPA system. Details are provided in the Appendix.

## Acknowledgements

We would like to thank Dr. James J Collins for the plasmid Para/lac and the *E.coli* K-12 MG1655*ΔlacI* and MG1655*ΔlacIΔaraCBAD* strain. We also thank Dr. Jeff Hasty for the microfluidic chip and setup. We thank Dr. Alexander Green for critical reading of the manuscript and great comments. FW was supported by American Heart Association Predoctoral Fellowship 15PRE25710303. YCL was supported by ARO under Grant No.W911NF-14-1-0504. This study was financially supported by National Science Foundation Grant DMS-1100309, American Heart Association grant 11BGIA7440101, and National Institutes of Health Grant GM106081 (to XW).

## Additional information

### Funding

| Funder | Grant reference number | Author |
| --- | --- | --- |
| American Heart Association | 15PRE25710303 | Fuqing Wu |
| Army Research Office | W911NF-14-1-0504 | Ying-Cheng Lai |

| National Science Foundation | DMS-1100309 | Xiao Wang |
| National Institutes of Health | GM106081 | Xiao Wang |
| American Heart Association | 11BGIA7440101 | Xiao Wang |

The funders had no role in study design, data collection and interpretation, or the decision to submit the work for publication.

## Author contributions

FW, Designed the research, Performed the experiments, Wrote the paper; R-QS, Designed the research, Developed the mathematical modeling, Wrote the paper; Y-CL, Wrote the paper; XW, Designed the research, Wrote the paper

## Author ORCIDs

Fuqing Wu, http://orcid.org/0000-0002-2820-3550
Ri-Qi Su, http://orcid.org/0000-0002-1311-7596
Xiao Wang, http://orcid.org/0000-0002-4056-0155

## Additional files

### Supplementary files

• Source code file 1. Flow cytometry data analysis file.

• Source code file 2. Bifurcation analysis in Figure 3.

• Source code file 3. Potential landscape calculation file.

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

## Appendix 1

## Quantitative analysis

### Mathematical modeling

#### Multimerization

Much of the gene regulation in our circuit involves multimerization of protein. These reactions can be described as:

$$2[luxR] \underset{k_-^u}{\overset{k_+^u}{\rightleftharpoons}} [luxR_2]$$

$$2[tetR] \underset{k_-^t}{\overset{k_+^t}{\rightleftharpoons}} [tetR_2]$$

$$2[LacI] \underset{k_-^l}{\overset{k_+^l}{\rightleftharpoons}} [LacI_2]$$

$$2[LacI_2] \underset{k_-^{l,d}}{\overset{k_+^{l,d}}{\rightleftharpoons}} [LacI_4]$$

$$2[AraC] \underset{k_-^a}{\overset{k_+^a}{\rightleftharpoons}} [AraC_2].$$

From these basic processes, we can calculate the concentrations for all the dimers and *LacI* tetramer (denoted using subscripts):

$$[luxR_2] = \frac{k_+^u}{k_-^u}[luxR]^2$$

$$[AraC_2] = \frac{k_+^a}{k_-^a}[AraC]^2$$

$$[tetR_2] = \frac{k_+^t}{k_-^t}[tetR]^2$$

$$[LacI_4] = \frac{k_{2+}^I}{k_{2-}^I}(\frac{k_+^I}{k_-^I})^2[LacI]^4.$$

#### Mathematical model for hybrid promoters

We use $P_{i,j}^A$ to denote promoter *Para/lac* with $i[AraC_2]$ and $j[LacI_4]$ binding, where $i = 0, 1$ and $j = 0, 1, 2$ since *Para/lac* has two binding sites for $[LacI_4]$. Similarly, we use $P_{i,j}^B$ to denote promoter *Plux/tet* with $i[luxR_2]$ and $j[tetR_2]$ binding, where both $i$ and $j$ can be 0 or 1. The association dynamics for these two hybrid promoters can be described as:

$$[AraC_2] + P_{0,j}^A \underset{S_-^a}{\overset{S_+^a}{\rightleftharpoons}} P_{1,j}^A, \quad j \in \{0,1,2\}$$

$$[LacI_4] + P_{i,0}^A \underset{S_-^l}{\overset{S_+^l}{\rightleftharpoons}} P_{i,1}^A, \quad i \in \{0,1\}$$

$$[LacI_4] + P_{i,1}^A \underset{S_-^l}{\overset{S_+^l}{\rightleftharpoons}} P_{i,2}^A, \quad i \in \{0,1\}$$

$$[luxR_2] + P_{0,j}^B \underset{S_-^u}{\overset{S_+^u}{\rightleftharpoons}} P_{1,j}^B, \quad j \in \{0,1\}$$

$$[tetR_2] + P_{i,0}^B \underset{S_-^t}{\overset{S_+^t}{\rightleftharpoons}} P_{i,1}^B, \quad i \in \{0,1\},$$

where $S_+$ and $S_-$ are protein-DNA association and disassociation rates, respectively, and the superscript represents corresponding repressor or activator, where $a$ represent rates for $[AraC_2]$, $l$ for $[LacI_4]$, $t$ for $[tetR_2]$ and $u$ for $[luxR_2]$. For simplicity, we have omitted the looping process of plasmid.

Using these binding/unbinding relationships, we can calculate the portion of $P_{1,0}^A$ and $P_{1,0}^B$ promoters. Both of them have maximal production rates compared to other promoter binding states, thus they are the dominant states among all possible binding states in determining circuit activities. From the first and second equations we can write down the ratio $R_{1,j}^A$ for promoter binded by $[AraC_2]$ (regardless of the binding of protein $[LacI_4]$) is:

$$R_{1,j}^A = \frac{P_{1,j}^A}{P_{1,j}^A + P_{0,j}^A} = \frac{\frac{S_+^a}{S_-^a}[AraC_2]}{1 + \frac{S_+^a}{S_-^a}[AraC_2]}. \tag{1}$$

We further analyze the ratios of promoters binded by $[LacI_4]$. Since there are two binding sites for $[LacI_4]$, their dynamics can be expressed as:

$$P_{i,0}^A \underset{S_-^l}{\overset{2S_+^l \cdot [LacI_4]}{\rightleftharpoons}} P_{i,1}^A \underset{2 \cdot S_-^l}{\overset{S_+^l \cdot [LacI_4]}{\rightleftharpoons}} P_{i,2}^A. \tag{2}$$

We can write down the dynamical functions for promoter $P_{i,0}^A$ and $P_{i,2}^A$ as:

$$\frac{dP_{i,0}^A}{dt} = S_-^l \cdot P_{i,1}^A - 2 \cdot S_+^l [LacI_4] \cdot P_{i,0}^A, \tag{3}$$

and

$$\frac{dP_{i,2}^A}{dt} = S_+^l [LacI_4] \cdot P_{i,1}^A - 2 \cdot S_-^l \cdot P_{i,2}^A. \tag{4}$$

So the stable ratio for $P_{i,0}^A$, or promoter without $[LacI_4]$ binding, is:

$$R_{i,0}^A = \frac{P_{i,0}^A}{P_{i,0}^A + P_{i,1}^A + P_{i,2}^A} = \frac{1}{\left(1 + \frac{S_+^l}{S_-^l}[LacI_4]\right)^2}. \tag{5}$$

Similarly, we can write down equations to describe $[luxR_2]$ and $[tetR_2]$ binding activities. Their dynamics can be expressed as:

$$\frac{dP_{1,j}^B}{dt} = S_+^u[luxR_2] \cdot P_{1,j}^B - S_-^u \cdot P_{0,j}^B,$$

$$\frac{dP_{i,0}^B}{dt} = S_-^t \cdot P_{i,1}^B - S_+^t[tetR_2] \cdot P_{i,0}^B,$$

and then solve the stable ratios $R_{1,j}^B$ and $R_{i,0}^B$ as:

$$R_{1,j}^B = \frac{\frac{S_+^u}{S_-^u}[luxR_2]}{1 + \frac{S_+^u}{S_-^u}[luxR_2]}, \tag{6}$$

$$R_{i,0}^B = \frac{1}{1 + \frac{S_+^t}{S_-^t}[tetR_2]}. \tag{7}$$

So at steady state, the ratios are controlled by association rates and the concentration of activator and inhibitors, as followed:

$$R_{1,0}^A = R_{1,j}^A \times R_{i,0}^A = \frac{\frac{S_+^a}{S_-^a}[AraC_2]}{1 + \frac{S_+^a}{S_-^a}[AraC_2]} \times \frac{1}{\left(1 + \frac{S_+^l}{S_-^l}[LacI_4]\right)^2}, \tag{8}$$

$$R_{1,0}^B = R_{1,j}^B \times R_{i,0}^B = \frac{\frac{S_+^u}{S_-^u}[luxR_2]}{1 + \frac{S_+^u}{S_-^u}[luxR_2]} \times \frac{1}{1 + \frac{S_+^t}{S_-^t}[tetR_2]}. \tag{9}$$

## Dosage response

Here, we model how the association rates and disassociation rates are affected by two inducers [Arabinose] and [AHL], as followed (*Stricker et al., 2008*):

$$S_a = \frac{S_+^a}{S_-^a} = C_{min}^a + [C_{max}^a - C_{min}^a] \cdot \frac{[Arabinose]^{n_a}}{[Arabinose]^{n_a} + K_a^{n_a}}, \tag{10}$$

$$S_u = \frac{S_+^u}{S_-^u} = C_{min}^u + [C_{max}^u - C_{min}^u] \cdot \frac{[AHL]^{n_u}}{[AHL]^{n_u} + K_u^{n_u}}, \tag{11}$$

and for the inhibition by aTc and IPTG as:

$$S_t = \frac{S_+^t}{S_-^t} = C_{min}^t + [C_{max}^t - C_{min}^t] \cdot \frac{K_t^{n_t}}{K_t^{n_t} + [aTc]^{n_t}}, \tag{12}$$

$$S_l = \frac{S_+^l}{S_-^l} = C_{min}^l + [C_{max}^l - C_{min}^l] \cdot \frac{K_l^{n_l}}{K_l^{n_l} + [IPTG]^{n_l}}. \tag{13}$$

## Leakage modeling

There are always leakage for promoters, so we further construct a leakage model to approximate these effects. $p$ is used to denote the basal production rate. Also, we use $a$ to denote the ratio of production rates between non-activator-binding promoters and activator-binding promoters, and $r$ to represent the ratio of production rates between inhibitor-binding promoters and non-inhibitor-binding promoters. Since the fluorescence strength can be directly measured and are proportional to gene expression levels, we use the concentration of mCherry protein, $[mC]$, to represent the activity of promoter *Para/lac* and the concentrations of *LacI* and *AraC*. Similarly, we use $[GFP]$ to represent the concentration of GFP protein and further the concentrations of *tetR* and *luxR*.

When leakage is introduced, we can write down the dynamical functions for MINPA as followed:

$$\frac{d[mC]}{dt} = p_m + A_{mC}\left[a_m(1-R_{1,j}^A)+R_{1,j}^A\right] \times \left[r_m(1-R_{j,0}^A)+R_{j,0}^A\right] - d \cdot [mC] \tag{14}$$

$$\frac{d[GFP]}{dt} = p_g + A_{GFP}\left[a_g(1-R_{1,j}^B)+R_{1,j}^B\right] \times \left[r_g(1-R_{j,0}^B)+R_{j,0}^B\right] - d \cdot [GFP]. \tag{15}$$

Here, $p_m$ and $p_g$ are basal leakage levels. $a_m$ and $r_m$ are reduced ratios of production rate for promoter *Para/lac*, while $a_g$ and $r_g$ are for promoter *Plux/tet*. $A_{mC}$ is the maximal production rates for *Para/lac* and $A_{GFP}$ is the maximal production rate for *Plux/tet*. We can further simplify this model to be:

$$\frac{d[mC]}{dt} = p_m + A'_{mC}\left[a'_m + R_{1,j}^A\right] \times \left[r'_m + R_{j,0}^A\right] - d \cdot [mC] \tag{16}$$

$$\frac{d[GFP]}{dt} = p_g + A'_{GFP}\left[a'_g + R_{1,j}^B\right] \times \left[r'_g + R_{j,0}^B\right] - d \cdot [GFP],$$

where $A'_{mC} = A_{mC} \cdot (1-a_m) \cdot (1-r_m)$, $A'_{GFP} = A_{GFP} \cdot (1-a_g) \cdot (1-r_g)$, $a'_m = \frac{a_m}{1-a_m}$, $a'_g = \frac{a_g}{1-a_g}$, $r'_m = \frac{r_m}{1-r_m}$ and $r'_g = \frac{r_g}{1-r_g}$.

## Dosage response for promoter tests

In order to fit the parameters for both promoter association and the dosage inductions, we construct and test the promoter expression rates in new constructed gene motifs as shown in *Figure 1—figure supplement 1B–C*.

As shown in *Figure 1D–E*, we perform dosage response experiments for *Para/lac* and *Plux/tet*. In each experiment, we grow the cells containing these gene circuits with different dosage combinations of inducer concentrations ([Arabinose] and [IPTG] for *Para/lac*, or [AHL] and [aTc] for *Plux/tet*) and then measure fluorescence strengths.

In both promoter setups, the concentrations of activators and inhibitors are controlled by constitutive promoters. According to the *Equation (14)* and *Equation (15)*, the stable fluorescence strengths mCherry and GFP are determined by inducer concentrations. The stable concentrations for [mC] and [GFP] are:

$$[mC]^*([Arabinose],[IPTG]) = \frac{1}{d}\left[p_m + A'_{mC}H_a([Arabinose]) \times H_l([IPTG])\right]$$

$$[GFP]^*([AHL],[aTc]) = \frac{1}{d}\left[p_g + A'_{GFP}H_u([AHL]) \times H_t([aTc])\right], \tag{17}$$

where $H_a([Arabinose]) = a'_m + R_{1,j}^A$, $H_l([IPTG]) = r'_m + R_{i,0}^A$, $H_u([AHL]) = a'_g + R_{1,j}^B$ and $H_t([aTc]) = r'_g + R_{i,0}^B$ are dosage response functions that need to be fitted. To isolate response functions for different dosages, we make the assumption that inducers function independently, thus we can isolate and then fit the Hill functions from the dosage response experiment.

Here we use principle component analysis (PCA) to analyze the dosage response data. PCA is a matrix decomposition method which project a matrix $M$ into a set of values of linearly uncorrelated variables called principal components, as $M = U\Sigma V^*$. Here $\Sigma$ is a diagonal matrix with non-negative real number on the diagonal, which represent the variability of the corresponding vectors in $U$ and $V$. Since we assume that the dosage response data are product of two single variable function, we can use the vectors corresponding to the largest variability in $U$ and $V$ as the dosage response functions.

We take the response functions for promoter *Para/lac* as an example to explain our fitting procedure. First we subtract different leakage levels of $p_m$ from the original data, and perform PCA to obtain response functions for activator and repressor, respectively. The decomposition vectors corresponding to the largest factors are considered as the response functions for $H_a([Arabinose])$ and $H_l([IPTG])$, which are plotted as circles in **Figure 1—figure supplement 1 D1 and D2** . We calculate the error between the experimental data and the model prediction from response functions, and then decide the optimal value for $p_m$. We can further determine the value for $p_g$ and the normalized $H_u([AHL])$ and $H_t([aTc])$ function in a similar approach (see circles in **Figure 1—figure supplement 1 D3 and D4**).

When the response functions are obtained, we can use the theoretical models for dosage response to fit the optimal leakage terms and the parameters in Hill functions (red solid lines in **Figure 1—figure supplement 1D**).

Using the models and these parameters, we can then determine the quantitative models for MINPA and the other circuits in **Figure 1—figure supplement 1A**.

## Network comparison by parameter searching

### Simplified models used in network comparison

We use the exhaustive parameter searching (**Faucon et al., 2014**; **Ma et al., 2009**) and simplified ODE models to quantitatively study the multistability of different circuits listed in **Figure 1—figure supplement 1A**. If a given circuit has larger possibility to be tuned by combination of inducers into multiple states, we consider such circuit has larger multistability likelihood.

First, we simplify the ODE model for circuit T15, the MINPA, and then try to derive simplified models for all the other circuits. The model for circuit T15 are:

$$\frac{d[x]}{dt} = p_0 + A'_{mC}\left(a'_0 + \frac{S_a[x]^4}{S_a[x]^4 + 1}\right) \times \left(r'_0 + \frac{1}{(S_l[y]^4 + 1)^2}\right) - d \cdot [x]$$

$$\frac{d[y]}{dt} = p_0 + A'_{GFP}\left(a'_0 + \frac{S_u[y]^4}{S_u[y]^4 + 1}\right) \times \left(r'_0 + \frac{1}{S_t[x]^4 + 1}\right) - d \cdot [y],$$

where we use $[x]$ to represent the activity of promoter *Para/lac*, and $[y]$ for promoter *Plux/tet*, respectively. The leakage levels, which are $p_0$, $a_0$ and $r_0$, are used to denote the leakage in promoter, activation and inhibition of the promoters. We assume that the leakage in promoter are small and symmetrical for both hybrid promoters in all circuits. For simplicity, we use the inducer affinity terms, e.g., $S_a$, $S_u$, $S_t$ and $S_l$, to represent the actual dosage induction effects. Generally, a larger affinity term for $S_a$ and $S_u$ represent higher dosage levels and also higher affinity rates. The meaning of $S_t$ and $S_l$ are on the contrary, when larger values represent lower dosage concentration.

When the corresponding genes are removed from circuit T15, we change the value of affinity to address the topology changes. If the corresponding self-activation is removed, e.g., in circuit T5, T7 and T13, there is no activator protein synthesized thus the affinity rates are ignorable. We then set $S_a$ and/or $S_u$ value as very small in these circuits. Similarly, if the repressor genes are removed, the affinity rate for $S_t$ and $S_l$ will become very small. We also argue that the leakage levels are unchanged since we assume the leakage terms are determined by the hybrid promoter part only and all circuits in **Figure 1—figure supplement 1A** share the same hybrid promoters. So we can use the same model to represent all synthetic gene circuits in **Figure 1—figure supplement 1A**, by setting the value of $S$ to a very small number to represent the elimination of certain link.

## Quantitative comparison

We use the same method (*Faucon et al., 2014*; *Ma et al., 2009*) to quantitatively evaluate the ability to achieve multistability for different circuits. Stronger ability to generate multistability is defined as easier to achieve multistability in arbitrary dosage combinations. So we set a reasonable ranges for the activation/repression strengthes $S$ of the existing links in the circuit to represent the variance of inducer concentrations, and randomly pick up a parameter combination from the given parameter space and calculate whether the parameter combination can generate more than two stable steady states (SSS). We choose $p_0 = 0.1$, $a_0 = 0.1$, $r_0 = 0.1$ and $d = 0.05$ in the numerical simulations. The ranges for activation/repression strength are $[0.3, 0.8]$. If the link is missing in certain circuit, we set the value for the corresponding activation/repression strengths as very small number, e.g., $0.01$. We repeat this procedure for 2000 times for all sub-networks in *Figure 1—figure supplement 1A* and then summarize the probability to find a suitable parameter combination and also the distribution for these parameters. As they are shown in *Figure 1—figure supplement 1H*, only circuits with two auto-activation links, which are $R^2A^2$(T15), $RA^2$(T11 and T14) and $A^2$(T10), can exhibit tristability (black solid lines) and quadrastability (red solid lines). Only parameter combinations for bistability can be found for circuit with one autoactivation link, i.e., $R^2A$(T7 and T13). The probabilities to find parameter combination for tristability and quadrastability are given in *Figure 1—figure supplement 1G*. We also show the parameter distributions for these four circuits in *Figure 1—figure supplement 1H*, while the distribution for fixed parameters in these circuits are not shown. We can also find that $R^2A^2$ has wider ranges of parameters for tristability and quadrastability than the other two circuits.

The scatter plots for all the four parameters in $R^2A^2$ that have tristability or quadrastability are shown in *Figure 1—figure supplement 1I*. The scatter plot between $S_a$ and $S_u$ are symmetrical, which suggests that the activation strengths have to be balanced to generate multistability. It is the same for the inhibition strength of $S_t$ and $S_l$. We can also find strong correlation between the activation strength and inhibition strength, especially when activation is strong. This is also the reason that the range for activation strength is enlarged. This scatter plots suggest that specific inhibition strength is required to generate multistability when the circuit has strong expression, however, when the activation is weak the multistability is not sensitive to inhibition strength.

## Hysteresis experiments and analysis for MINPA

We perform a series of hysteresis experiments and bifurcation analysis, including single inducer and dual inducer, to fully explore the multiple stability behaviors of MINPA.

### Single dosage induction

Before performing bifurcation analysis, we verify and adjust the quantitative model *Equation (16)*, which is built on multimerization and dosage response curves using the transition dosage in hysteresis experiments for Arabinose and AHL in *Figure 3—figure supplement 2I* and *Figure 3—figure supplement 2J*. We also find that, the system can maintain both LL and HH states as they are shown in panel (C1$_{LL}$) in *Figure 3C* and panel (C1$_{HH}$) in *Figure 3D*. From these experimental results and modeling analysis, we can hypothesize that the MINPA system is quadrastable when no inducer is added.

The bifurcation analysis are performed using Matlab and bifurcation analysis package matcont. Firstly, we exhaustively search all steady states in the initial conditions. We divide the phase space into $10 \times 10$ grids. Within each grid, we solve *Equation (16)* for the steady states with a random of initial solution located within this grid. Secondly, we initialize from each steady state and using parameter continuation package provided by matcont to go through this branch when the parameter increases. Similarly, we explore all branches

initialized from all steady states. Finally, we use Jacobian method to study the stability of all the branch points and then mark the SSS to blue and USS to red.

We use the logarithm of inducer concentrations to perform bifurcation analysis. In *Figure 3—figure supplement 2A*, initially there are 4 SSSs when all the concentrations of inducers are zero (marked as blue solid lines). When Arabinose increases, the LL state vanishes and become mCherry state, and the GFP state will also become HH state. The HH and mCherry states remain stable even at very high dosage of Arabinose. The bifurcation suggests that we can drive the system from LL state to mCherry state using Arabinose. Also the increase of Arabinose can help the system overcome the barrier between GFP and HH state.

On the contrary to the bifurcation of Arabinose, the bifurcation of AHL suggests that the LL state will not vanish when the concentration of AHL increases, as it is shown in *Figure 3— figure supplement 2B*. Only the mCherry state will vanish so AHL can be used to drive the mCherry state to HH state. Meanwhile, the GFP state and HH state remain stable. The branches of USS will change as AHL increases, and it suggests that AHL can be used to modify the stability of associated SSSs.

In *Figure 3—figure supplement 2C–D* we can see that, all branches of SSSs will not disappear when the concentration of inducers of aTc and IPTG increase. So changing the concentration of aTc or IPTG solely cannot drive the system from one stable state to the other one. Similarly to AHL, the USS branches will change their position thus the stability of SSSs will change accordingly.

From these single induction experiment, we can see there are many limitations in driving the initial LL state to all the other three states. The Arabinose can drive the system out of low-mc states. However, AHL, aTc and IPTG can not drive the system out of LL state solely by themselves. They can only change the stability of LL state. For these reasons, we need to combine different dosage to efficiently change the landscape of SSSs and further control the differentiation.

## Dual induction

We design a dual induction approach to study the bifurcation behaviors in two dimensional parameter space. Given two changing parameters $[D_1]$ and $[D_2]$ and GFP and mCherry, it will be challenging to visualize and interpret results in four dimensions. Here we use a hybrid bifurcation technique, in which the concentration ratio $\beta = [D_1]/[D_2]$ between two changing parameters is fixed, to explore the parameter space along different directions. The hybrid bifurcation returns to the conventional one-dimensional bifurcation when $\beta = 0$ or $\beta = \infty$.

In *Figure 3C*, we can see that, by choosing the dual inducers as Arabinose and AHL and $\beta = 10$, the MINPA system initialized from LL state will transit to mCherry and then HH state. In low mix dosage concentration, the system will stay at LL state and start to transit to either mCherry or GFP state when the mix concentration increases. Finally, all of the systems will rest in HH state. Different value of $\beta$ can have slightly different transition behaviors, as they are shown in *Figure 3—figure supplement 2E–H*. When the value of $\beta$ increases from 0.1 to 100, the system would prefer to transit to the GFP state more than the mCherry state. We decide to choose $\beta = 10$ because the MINPA will have wider quadrastable region than the others.

## Sequential induction

We study the effect of sequential inductions using bifurcation analysis and stochastic simulations. Compared to the hybrid bifurcation which apply the inducers at the same time, the sequential induction applies one inducer first and then the other inducer after a duration of $T$. The essence of sequential induction is that the induction effect depends on the SSS which the system stays in. For example, if the system is already at high-mC states, the

inhibition from mCherry to GFP would create extra obstacle for the system to transit from GFP low to GFP high state. For this reason, it will require higher concentration of inducers to drive the system from mCherry state to HH state than the process from LL state to GFP state.

In deterministic systems, as it is shown in *Figure 2A*, different induction sequences may drive the gene network to different terminal states. We perform detailed bifurcation analysis and find that the interaction between inducer one and inducer two on their bistable regions can lead to different terminal states. The two dimensional phase diagram is show in *Figure 2A*. When the concentration of inducer one increases, the bistable region for inducer two will become broader. Also, its lower boundary increases, which suggest that the toggle will need more inducer two to maintain its high GFP response state. It is similar for the increase of inducer two. We assume that the system initialize at high GFP state and neither concentration of inducer one nor inducer two can cause state transition to low GFP state. If inducer two is introduced first, the bistable region for inducer one will change and the required concentration of inducer two to induce state transition will become smaller because the lower boundary of its bistable region increase. So a state transition will happen because of the sequence of induction. However, if sequence is permuted or two inducers are applied simultaneously, the system remain in high state. We can also learn from the two dimensional bifurcation diagram that it will have larger chance to discover the state transition boundary, comparing to using the hybrid induction method alone (the diagonal induction curve).

We now show how to carefully perform experiment to observe the sequential induction effects based on numerical calculations. First we constructed *E. Coli* plasmids which contain mutual inhibitory circuit and can be induced by aTc and IPTG, as it is shown in *Figure 2—figure supplement 1A*. Then we chose two dosages of IPTG, $D_I L$ and $D_I H$ so that the system could be bistable for aTc under these two different IPTG concentrations. From previous numerical studies, we found that as long as the right boundary of bistable region, $B_a H$ under high IPTG was shifted to the right of the one $B_a L$ under low IPTG, the sequential effect can be observed by setting the high aTc concentration $D_a H$ between $[B_a L, B_a H]$. The low aTc concentration $D_a L$ shall be less than $B_a L$. In our experiment shown in *Figure 2—figure supplement 1B*, we chose $D_I H$ as $8 \times 10^{-5} M$, $D_I L$ as 0, $D_a H$ as 100 ng/ml, and $D_a L$ as 0.

Different sequences of induction provide more alternatives to explore the parameter space from different directions and can help to generate different population distributions in stochastic systems with appropriate low and high inducer concentrations, given finite amount of time. We can also use this property to design new cell differentiation protocols using this method. The transition rate between two neighboring SSSs depends on the barrier height between them. For example, in order to induce the cells from LL state to GFP state, we need to jointly add aTc and AHL to lower the barrier height between LL and GFP state. Some barrier can be very high thus it is impossible for the system to overcome the barrier under given inducers, e.g., the one between mCherry state to HH state. We can also use the sequential induction to compare the multistability in varying circuits in *Figure 1—figure supplement 1A*.

## Construct quasi-potential attractor landscape

Because of the nonlinearity of gene regulation and inducer response functions, the constructed gene networks is highly dissipative thus a conventional potential can not be defined (*Wang et al., 2011*). For this reason, we construct a quasi-potential landscape to describe the state transition barriers and stability of all SSSs. First, we made the assumption that all the gene network described by these dynamical functions will approach stationary state, where the densities of final state is a constant. We can define the pseudo potential from the stationary densities $P(x)$, and can be defined as $E(x) = -logP(x)$. In the conventional canonical dynamical system, the descending direction of a landscape is always pointing to the minimum of the landscape, however, it is not same in this dissipative system. Also,

because of the high dimensional state space and multiple attractor geometry, the convergence to stationary distribution is always slow under the transitional random walk algorithm. In order to address this obstacle, we applied the so-called weight ensemble random walk method to speed the convergence rates (**Kromer et al., 2013**).

We describe in detail the procedure of calculating the stable density distribution under the dynamical function **Equation 16** as follow. First, we need to determine the time step $\tau$ and the space resolution $\Delta x$ according to reference (**Kromer et al., 2013**) to ensure convergence. When the noise strength is $D$, the time step of simulation $\tau$ shall satisfy: $\tau < \frac{4D}{max f^2(\mathbf{X})}$, and $f(\mathbf{X})$ is the velocity field. Also, $\Delta x$ shall satisfy $\Delta x \leq f(\mathbf{X})\tau$. Then we can discretize the state space, or possible ranges of protein concentration, into $M \times M$ lattices as $M = 1/\Delta x$. We will use the densities of each lattice $P_{m,n}$ to approximate densities $P(x)$ in the state space. The initial probability $P_{m,n}(0)$ of all gird points are set to be uniform, as $P_{m,n}(0) = \frac{1}{M \times M}$. When the simulation is long enough, the initial distribution would not affect the final stationary distribution.

At each step of duration $\tau$, we randomly place $N$ walkers within each grid $[m, n]$, no matter how small its probability is. Each of these walkers carries equal weight $q_{m,n}^i(t) = P_{m,n}(t)/N$ and start evloving from its initial position under the system dynamics and noise, and 'transport' to nearby grids. Its evolution can be simulated using any numerical integration of **Equation 16**. Here we choose the stochastic version of second order Runge-Kutta algorithm, or the Heun algorithm. When all of the walkers' position have been updated, the new probabilities for each grid is:

$$P_{m,n}(t + \tau) = \sum_{k \in G_{m,n}} q_{m',n'}^k(t), \tag{18}$$

where $q_{m',n'}^k(t)$ is the probability that carried by a walker initialized at grid $[m', n']$ and fell into grid $G_{m,n}$. At next time step, probabilities carried by another $N$ new walkers will be updated according to the new updated probability. This procedure repeats until the probability distribution $P_{m,n}(t)$ becomes stationary. The landscape can be calculated from $E_{m,n} = -log P_{m,n}$.

Using potential landscape we can estimate the relative stability of different SSSs and the hardness, or the barrier heights for the transition between two SSSs.

## Quadrastability induction in microfluidic device

Microfluidics coupled with time-lapse imaging was employed to visualize the state transitions at the single-cell level. Two media were prepared: one with inducers (Arabinose, AHL and aTc) and the other without. After cells were loaded into the trap (1 to 5 cells for a trap), the device was heated up to 37 degree and cells were supplied with LB media without inducers for 6 hr. Sulforhodamine was added as a dye to monitor nutrient transport. Then, the supplied media was switched to the media added with Arabinose, AHL and aTc for another 18 hr. Media switching was controlled by adjusting the heights of the medium syringes relative to one another.

However, cells treated with the three inducers demonstrate symptoms of significant stress and cell death, presumably due to photo toxicity compounded with flow-induced sheer stress and other mechanical stresses in the microenvironment (**Kohles et al., 2009**; **Shen et al., 2014**; **Shemesh et al., 2015**). Lower concentrations of three inducer combinations were also tested but yield no significant improvement of cells viability.

Since the logic of emergence of quadrastability is enhancing the two positive feedbacks of MINPA through adding inducers Arabinose, AHL and aTc, quadrastability could also be achieved through weakening the mutual inhibition using IPTG and aTc. Depending on the

basal expression of two hybrid promoters, IPTG and aTc can promote GFP and mCherry expression to a limited extent, which in turn attenuates fluorescent proteins toxicity. Hence, we tried to use IPTG and aTc to induce quadrastability instead of the three inducers tried in flow cytometry. Experimental result showed that the initial low-low state cells could differentiate into GFP, mCherry and high-high state cells with $2 \times 10^{-4}$ M IPTG and 200 ng/ml aTc induction (*Figure 4—figure supplement 1D* and *Appendix 1—video 1*). It is interesting that the trajectory for many cells were from GFP to high-high to mCherry state. Altogether, this result further verified the MINPA has the potential to generate quadrastability in living cells.

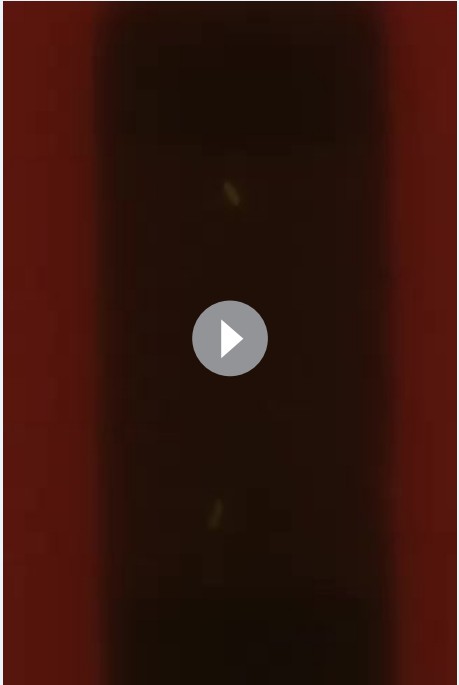

**Appendix 1—Video 1.** A time-lapse movie growing in the microfluidic chip for 24 hr. Time course of the cells growing and fluorescence state change with 2*10$^{-4}$ M IPTG and 200 ng/ml aTc induction in the trap. The red flow is medium without inducer for 6 hr, and then cells switch to medium with inducers for 18 hr. Magnification: 40x.

## Cell growth rates under inductions

Cells with the MINPA plasmid were cultured overnight at 37 degree and diluted into fresh media with corresponding inducers at 1:100 ratio (O.D. $\approx 0.066$). The four individual inducers are Ara ($2.5 \times 10^{-5}$ m/v), AHL ($1 \times 10^{-5}$ M), aTc (200 ng/ml), IPTG ($1 \times 10^{-4}$ M), and inducer combinations: AHL and aTc, AHL, aTc, and Ara. Cellular growth rates were measured by using $200 \mu$L cultures in a 96-well plate with absorbance at 600 nm on a plate reader (BioTek, USA). Three replicates were tested for each condition.

Compared to AHL or Arabinose individually, aTc addition (aTc, AHL + aTc, and Ara + AHL + aTc) influenced cells growth and increased the lag phase for about 2.5 hours (*Appendix 1—figure 1*). But at about 13 hr, the growth rates are almost the same. Since the timescale in our experiments is much longer ($\geq$ 24 h) than 13 hr, we reason the effect of inducers on the cellular growth rates would not change interpretations of experimental observations.

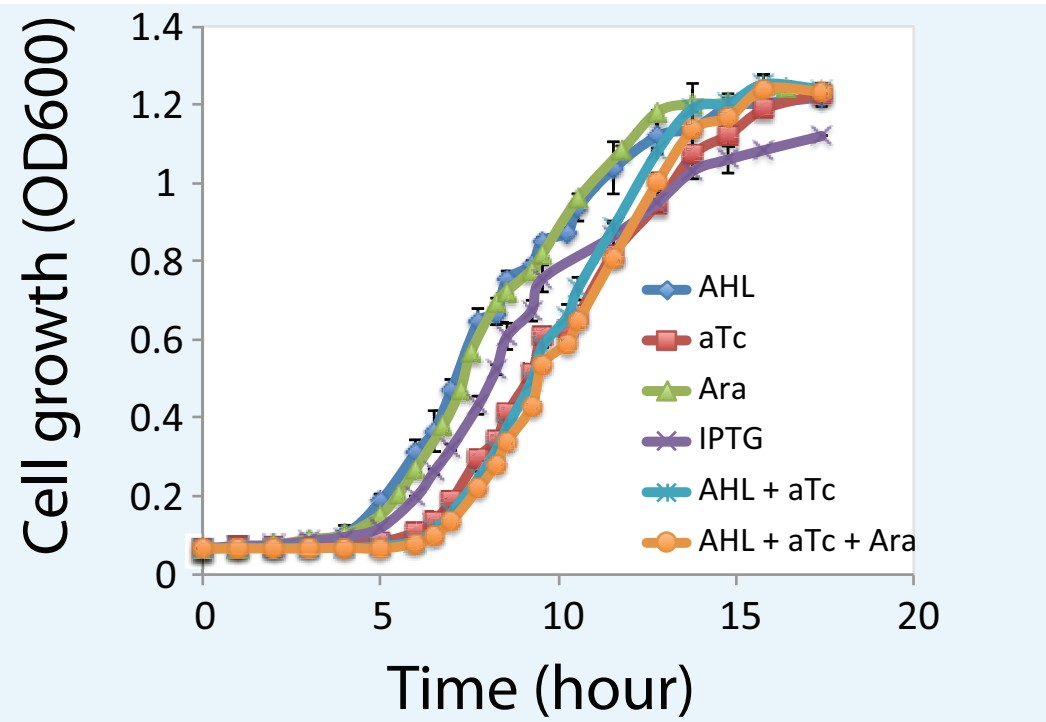

**Appendix 1—figure 1.** Cell growth rates under each inducer and inducer combinations. Growth curves for the cells under the four individual inducers: Ara; AHL; aTc; IPTG, and inducer combinations: AHL and aTc; AHL, aTc, and Ara. Ara: $2.5*10^{-5}$ m/v; AHL: $1*10^{-5}$ M, aTc: 200 ng/ml, IPTG: $1*10^{-4}$ M. Cells under induction with aTc has a longer lag phase (~2.5 hr), and all the samples reached stationary phase after ~13 hr. Data indicate mean±SD of three independent replicates.

