## [Decision Letter]

Thank you for submitting your article "Engineering of a synthetic quadrastable gene network to approach Waddington landscape and cell fate determination" for consideration by *eLife*. Your article has been favorably evaluated by Naama Barkai (Senior Editor) and three reviewers, one of whom, Wenying Shou (Reviewer #1), is a member of our Board of Reviewing Editors. The following individual involved in review of your submission has agreed to reveal their identity: Gabor Balazsi (Reviewer #2).

The reviewers have discussed the reviews with one another and the Reviewing Editor has drafted this decision to help you prepare a revised submission.

Summary:

To quantitatively and experimentally understand Waddington's landscape analogy commonly used to depict multiple cell fates, authors constructed a genetic network called MINPA (mutual inhibition and positive autoactivation). By analyzing various sub-networks of MINPA and the full MINPA network, authors find that the full MINPA is capable of achieving the most fate states of mCherry/GFP expression patterns. A total of four states can be achieved, with low or high mCherry and GFP. Aided by modeling, the authors charted how the "energy" landscape of cell fates changes in the presence of various chemicals, and was able to "guide" cells to transit from one state to another by applying chemicals at the correct orders.

All three reviewers thought that your work was important and interesting. However, there are issues that will need to be clarified and addressed before acceptance. Details are provided in the reviewers' comments below. Please note that depending on your response, reviewers might conclude that additional major experiments could be necessary. To help us assess the likelihood of such an outcome, it would help if you responded to the critiques with a plan of action and a timetable for completion of the work recommended by the reviewers. The editor and reviewers will assess your response and provide a recommendation to assist you in preparing a revised submission.

*Reviewer #1:*

I enjoyed reading it as someone with an informal interest in landscape.

Figure 3: Not clear to me that at C2, the mCherry state (red dot) has low GFP. Need to make the 3-dimensional illustration more clear.

Figure 3: How long can the HH state be maintained? Does your model also predict the time scale of decay?

Needs to better describe how "potential" in Figure 4 is calculated.

*Reviewer #2:*

This is an important and interesting manuscript that investigates experimentally the possibility of multistability in a library of two-gene networks with increasingly complex connectivity consisting of mutual inhibition and positive autoregulation. This topology has counterparts in mammalian cell fate-regulatory networks. The library consists of several synthetic networks, with a number of regulatory links that systematically increases from 2 to 4. Analyzing the dynamics of these networks reveals that the one with all 4 links present has the highest chance for multistability, with more than 2 stable steady states for a broad range of parameters. These findings are experimentally validated with clever sequential induction experiments and hysteresis using 4 different inducers.

The manuscript represents an important step forward in synthetic biology and beyond, having relevance to many cell fate determination networks, including in higher organisms. Nonetheless, some details are clarifications are still needed.

Therefore, I recommend publication once the following comments can be addressed:

1) Synthetic biology often measures its progress by the number of genes in synthetic gene circuits. However, this view ignores the fact that biological complexity does not really correlate with gene number. In fact, biological complexity correlates with the number of regulatory connections. Some plants have more genes than us, humans. In fact, a 10-gene network may be much simpler from a dynamical or an information-processing perspective than a two-gene network – provided that the latter has more complex connectivity. The arguments on network complexity resulting from links rather than nodes may be a nice addition to the Discussion. See Science 292(5520):1315-6 (2001).

2) From the Methods it seems like these networks are on high-copy plasmids rather than integrated. What is the effect of plasmid copy number variation on the results? Could the plasmid be lost from some cells, generating the impression of multistability? Flow cytometry at multiple time points will provide at least a partial answer.

3) The sequential induction method is quite interesting and important for revealing multistability. However, its application to the T15 network (or other networks) could be explained better. I guess the sequence in which inducers are added determines how the network's dynamics changes from monostable to multistable – but the details may be important. If bistable, tristable, quadristable regions are small in the 4-dimensional space then the goal is to "hit" these regions as inducers are sequentially added. While this is shown for the toggle switch, it is unclear how was this done for the more complex networks. This should be much more clearly described, especially for T15 and possibly for some simpler networks. Overall, going from the toggle switch to T15 and even to simpler networks is a big step, which requires more explanation and investigation. In addition, to toggle switch would be worth adding to the experimentally measured networks as control, considering that its behavior is computationally predictable. Do the experiments validate the predictions in Figure 2 for the toggle switch?

*Reviewer #3:*

In this work, the authors quantified the cell fate decision making landscape through engineering the synthetic circuits. They have identified multiple states. They also considered the cell fate decision at different conditions. The results are interesting and I recommend its publication after revisions upon the following comments.

1) Recently, the theoretical advances have been able to identify the landscape and flux as the driving force of the global dynamics of the biological circuits (Proc. Natl. Acad. Sci. USA, 105: 12271-12276. (2008).). The quantification of the Waddington landscape has been achieved and is directly related to the underlying gene regulatory network for cell fate decision of stem cell differentiation and development (Proc. Natl. Acad. Sci. USA. 108(20):8257-8262(2011).). These should be reflected in the revision.

2) Furthermore, the multiple states have been predicted beyond the bistable states/switches without and with the epigenetic effects (Proc. Natl. Acad. Sci. USA. 108(20):8257-8262(2011); J. Phys. Chem. B, 115, 1254 (2011); Sci. Rep., 2,550 (2012); J R Soc Interface 10: 20130787 (2013); PLoS ONE. 9(8): e105216 (2014); Advances in Physics, 64:1, 1-137. (2015).). These should be reflected in the revision.

3) The authors should explain more clearly the physical and biological origins of the quadrastable states. Why the triple stable states are less likely than the quadrastable states? What is the origin of the LL state?

---

## [Author Response]

*[…] Reviewer #1:*

*I enjoyed reading it as someone with an informal interest in landscape.*

*Figure 3: Not clear to me that at C2, the mCherry state (red dot) has low GFP. Need to make the 3-dimensional illustration more clear.*

We thank the reviewer for this great comment. To make the illustration clearer, especially for GFP at C2, we rotated the diagram and got a better view of the states at C2, which is added as Figure 3—figure supplement 1.

*Figure 3: How long can the HH state be maintained? Does your model also predict the time scale of decay?*

In our experiments, we found the HH state is very stable. When we tried to use AHL and Arabinose to induce MINPA, the HH state could be maintained up to 24 hours after induction. In the hysteresis results (Figure 3 and Figure 3—figure supplement 2), the HH state cells were washed and resuspended in fresh media with Arabinose and AHL, and the state is stable for at least 24 hours.

In our deterministic and stochastic models, protein degradation is the main factor determining time scales of state transitions. Once the system reaches its stable steady states, it will not change states until new induction is applied. Therefore, our model could qualitatively represent speed of state transitions by choosing appropriate protein degradation rate.

*Needs to better describe how "potential" in Figure 4 is calculated.*

We thank the reviewer for prompting us to improve the manuscript. As suggested, we summarized the method to calculate the “potential” in the revised main text (subsection “Experimental demonstration of model-guided quadrastability of MINPA”, first paragraph), and more details can be found in the Appendix (“Construct Quasi-potential Attractor Landscape”).

*Reviewer #2:*

*[…] 1) Synthetic biology often measures its progress by the number of genes in synthetic gene circuits. However, this view ignores the fact that biological complexity does not really correlate with gene number. In fact, biological complexity correlates with the number of regulatory connections. Some plants have more genes than us, humans. In fact, a 10-gene network may be much simpler from a dynamical or an information-processing perspective than a two-gene network – provided that the latter has more complex connectivity. The arguments on network complexity resulting from links rather than nodes may be a nice addition to the Discussion. See Science 292(5520):1315-6 (2001).*

We thank the reviewer for this great suggestion. We have added it into the second paragraph of our Discussion section.

*2) From the Methods it seems like these networks are on high-copy plasmids rather than integrated. What is the effect of plasmid copy number variation on the results? Could the plasmid be lost from some cells, generating the impression of multistability? Flow cytometry at multiple time points will provide at least a partial answer.*

We thank the reviewer for this question. We constructed these networks on high-copy plasmids. It is true that the plasmid copy number in different cells harboring the same network and backbone may not be the same, resulting in varied gene expression levels within a cell population. Moreover, stochasticity in gene expression arising from fluctuations in transcription and translation would also lead to different levels of gene expressions. We think these factors are two of the most important ones causing noisy gene expression. Experimentally, the consequence of plasmid copy number and gene expression stochasticity is reflected on the size of each population. For example, the Low-Low state cells in Figure 3 is a population with green fluorescence ranging from 3*10^2^ to 4*10^3^ (a.u.), and GFP state cells from 10^4^ to 2*10^5^ (a.u.). If all the cells have the non-fluctuating gene expression, we would expect a much smaller cloud on the 2-D fluorescence plots.

As the reviewer commented, it is possible that the plasmid may be lost from some cells, especially for long-term culture in which the active antibiotic concentration decreases over time. The cells lost their plasmid would have no fluorescent protein expression in a long time course, and might be only mixed/hidden with the low-low state cells. So it would not influence cells in the other three states (GFP, mCherry, and high-high states). A time-course induction with AHL, aTc first and then Arabinose (Left route in Figure 4) were provided in the revision (Figure 4—figure supplement 1), from which the low-low state cells has a percentage from 79.1% (0 h after inducer II addition) to 21.2% (12 h), to 24.6% (24 h), to 30.6% (36 h). If the multistability were generated by the plasmid lost, then it would not be stable for the four populations till 36 h. Thus, we think the plasmid lost may increase the portion of low-low state cells for long-time cultures, but would not generate the four populations.

*3) The sequential induction method is quite interesting and important for revealing multistability. However, its application to the T15 network (or other networks) could be explained better. I guess the sequence in which inducers are added determines how the network's dynamics changes from monostable to multistable – but the details may be important. If bistable, tristable, quadristable regions are small in the 4-dimensional space then the goal is to "hit" these regions as inducers are sequentially added. While this is shown for the toggle switch, it is unclear how was this done for the more complex networks. This should be much more clearly described, especially for T15 and possibly for some simpler networks. Overall, going from the toggle switch to T15 and even to simpler networks is a big step, which requires more explanation and investigation. In addition, to toggle switch would be worth adding to the experimentally measured networks as control, considering that its behavior is computationally predictable. Do the experiments validate the predictions in Figure 2 for the toggle switch?*

We thank the reviewer for this great comment. We think sequential induction is a good strategy to quickly evaluate multistable potentials and screen out systems harboring the most interesting dynamics. We added more detail about the sequential induction strategy and the application to T15 and other networks in the revised manuscript (subsection “Systematical multistability evaluation of MINPA and its sub-networks”, fourth paragraph), and Appendix (“Sequential Induction”).

To validate the theoretical analysis for sequential induction in Figure 2, we constructed a synthetic toggle switch circuit and tested with sequential inductions. We first employed IPTG to induce the circuit for 5 h, and then aTc was added. Time course results showed that cells stayed at low-GFP state till 24 hours (Figure 2—figure supplement 1). However, cells induced with aTc first, and then IPTG mainly stayed at high-GFP state, another stable steady state under this condition. Simultaneous aTc and IPTG induction produced similar cell distributions. These results show that sequential induction can be used as a strategy to quickly explore a multistable potential landscape for complex non-equilibrium systems. Details can be found in the third paragraph of the subsection “Systematical multistability evaluation of MINPA and its sub-networks”, Figure 2—figure supplement 1, and subsection “Strains, Media, and Chemicals”

*Reviewer #3:*

*[…] 1) Recently, the theoretical advances have been able to identify the landscape and flux as the driving force of the global dynamics of the biological circuits (Proc. Natl. Acad. Sci. USA, 105: 12271-12276. (2008).). The quantification of the Waddington landscape has been achieved and is directly related to the underlying gene regulatory network for cell fate decision of stem cell differentiation and development (Proc. Natl. Acad. Sci. USA. 108(20):8257-8262(2011).). These should be reflected in the revision.*

We thank the reviewer for the great suggestions. We have added the discussion and corresponding references in the revision (Introduction, second paragraph; and Discussion, first paragraph).

*2) Furthermore, the multiple states have been predicted beyond the bistable states/switches without and with the epigenetic effects (Proc. Natl. Acad. Sci. USA. 108(20):8257-8262(2011); J. Phys. Chem. B, 115, 1254 (2011); Sci. Rep., 2,550 (2012); J R Soc Interface 10: 20130787 (2013); PLoS ONE. 9(8): e105216 (2014); Advances in Physics, 64:1, 1-137. (2015).). These should be reflected in the revision.*

We thank the reviewer for the great suggestions. We have added the discussion and corresponding references in the revision (Introduction, second paragraph).

*3) The authors should explain more clearly the physical and biological origins of the quadrastable states. Why the triple stable states are less likely than the quadrastable states? What is the origin of the LL state?*

We thank the reviewer for the great question. The quadrastability presented in this work comes from coherent regulations in GFP and mCherry expressions. Experimentally, we used two hybrid promoters *Para/lac* and *Plux/tet* to drive the regulations. Each of the two promoters has two protein binding sites. AraC activates *Para/lac* in the presence of arabinose while LacI can bind and block its transcription. LuxR activates *Plux/tet* with induction of AHL while TetR can repress its transcription. Hence, each of the promoters has two possible states: ON and OFF. The interlocked circuit design of MINPA enables it has four possible expression levels in a cell population: *Para/lac* ON and *Plux/tet* OFF, *Para/lac* OFF and *Plux/tet* ON, *Para/lac* OFF and *Plux/tet* OFF, and *Para/lac* ON and *Plux/tet* ON. Since GFP and mCherry are reporters of the two promoters, so they could represent four stable states: GFP, mCherry, low-GFP and low-mCherry, and high-GFP and high-mCherry states.

Physically, the quadrastable states stem from the MINPA design, which is described as ordinary differentiation equations. Our mathematical analysis revealed that there are four stable steady states in the MINPA system. We didn’t compare the emergence probability between tristability and quadrastability. But experimentally, it is shown that tri-modality is easier to emerge, such as in the case of induction of MINPA with Arabinose and AHL (Figure 3 (C2_LL_, C3_LL_)). However, quadra-modality can only occur with triple inductions (Arabinose, AHL and aTc, Figure 4, and Figure 4−figure supplement 1E).

We also compared our model with previous studies describing the same topology (Proc. Natl. Acad. Sci. USA. 108(20):8257-8262(2011); PLoS ONE. 9(8): e105216 (2014)), we found the main difference is the description of the relationship between positive autoregulation and negative feedback. Previous studies described it as a “plus” (which means the two events are independent and additive), however, we described it as a “multiply” (which means the two events are dependent and multiplicative). Specifically, *Para/lac* activation depends on the relative concentrations of AraC (from node X) and LacI (from node Y) in the cell. LacI binding to the promoter would block RNA polymerase progression and hence repress transcription. Therefore, the LL state origins from the high effective concentrations of LacI and TetR in the cell. We believe incorporating mutual dependence of two binding sites of the hybrid promoter is more appropriate for the MINPA system described in this work, although the “additive” description of hybrid promoter could be suitable for other systems with different regulatory mechanisms.